# Effects of diets on risks of cancer and the mediating role of metabolites

Yi Fan [1,2,4], Chanchan Hu [1,4], Xiaoxu Xie [1,4], Yanfeng Weng [1,4], Chen Chen [1], Zhaokun Wang [2], Xueqiong He [2], Dongxia Jiang [2], Shaodan Huang [2,3] ✉, Zhijian Hu [1] ✉ & Fengqiong Liu [1] ✉

Research on the association between dietary adherence and cancer risk is limited, particularly concerning overall cancer risk and its underlying mechanisms. Using the UK Biobank data, we prospectively investigate the associations between adherence to a Mediterranean diet (MedDiet) or a Mediterranean-Dietary Approaches to Stop Hypertension Diet Intervention for Neurodegenerative Delay diet (MINDDiet) and the risk of overall and 22 specific cancers, as well as the mediating effects of metabolites. Here we show significant negative associations of MedDiet and MINDDiet adherence with overall cancer risk. These associations remain robust across 14 and 13 specific cancers, respectively. Then, a sequential analysis, incorporating Cox regression, elastic net and gradient boost models, identify 10 metabolites associated with overall cancer risk. Mediation results indicate that these metabolites play a crucial role in the association between adherence to a MedDiet or a MIND-Diet and cancer risk, independently and cumulatively. These findings deepen our understanding of the intricate connections between diet, metabolites, and cancer development.

Cancer is one of the most important global public health concerns since it is responsible for a significant portion of global morbidity and mortality[1]. In 2020, it was estimated that approximately 18.1 million new cases of cancer, and nearly 10 million cancer-related deaths occurred worldwide[2]. The 3rd Expert Report of the World Cancer Research Fund (WCRF) expressed a consensus that various nutritional factors may have a "probable" causal, or protective association with cancer risk[3]. This consensus is supported by research conducted across various nations, including the United Kingdom[4], Brazil[5], Japan[6], the United States[7], and China[8].

A Mediterranean diet (MedDiet) and a Mediterranean-Dietary Approaches to Stop Hypertension (DASH) Diet Intervention for Neurodegenerative Delay diet (MINDDiet) are dietary patterns which have been widely reported to be protective against adverse health effects,

including cancer[9,10]. The MedDiet is distinguished by its high consumption of vegetables and fruits, regular intake of fish and seafood, and moderate intake of red wine[11]. It has been widely recognized as one of the healthiest dietary patterns globally. Multiple studies have found that adherence to a MedDiet can reduce the risk and mortality of metabolic diseases[12–14]. The MINDDiet was inspired by the renowned MedDiet but has been modified based on the most compelling evidence from human epidemiological and animal studies in the diet-dementia field[15]. The MINDDiet places a particular emphasis on natural plant-based foods while limiting the intake of animal and high saturated fat foods. Diverging from MedDiet, the MINDDiet distinctively encourages only the specific consumption of green leafy vegetables and berries, while not explicitly addressing the intake of other types of fruits. Most of the existing MINDDiet studies have concentrated on its

[1]Department of Epidemiology and Health Statistics, Fujian Provincial Key Laboratory of Environment Factors and Cancer, School of Public Health, Fujian Medical University, Fuzhou, China. [2]Department of Occupational and Environmental Health Sciences, School of Public Health, Peking University, Beijing, China. [3]Key Laboratory of Epidemiology of Major Diseases (Peking University), Ministry of Education, School of Public Health, Peking University, Beijing, China. [4]These authors contributed equally: Yi Fan, Chanchan Hu, Xiaoxu Xie, Yanfeng Weng. ✉e-mail: shhuang@bjmu.edu.cn; huzhijian@fjmu.edu.cn; lfq@fjmu.edu.cn

impact on neurodegenerative disease. These studies have suggested that adherence to a MINDDiet is associated with better cognitive performance[15], physical function[16], and mental state[17], as well as reduced risk of developing Alzheimer's disease (AD)[18] and cardiovascular disease (CVD)[19].

However, current studies on the preventive effects of a MedDiet and MINDDiet on cancer risk, still face several limitations. First, the conclusions of the current research are still open to further discussion. For MedDiet, findings about its associations with cancer are still inconclusive. According to prospective cohort studies conducted by Lavalette et al.[20] and Bodén et al.[21] in France and Sweden respectively, there was no significant association between a MedDiet and overall cancer risk; however, a recent meta-analysis encompassing 117 studies provided compelling evidence indicating a reduction in overall cancer risk with greater adherence to a MedDiet[22]. For the MINDDiet, the association with cancer has primarily been examined in a few case-control studies, suggesting its inverse associations with the risk of some specific cancers, e.g., breast cancer[23–25] and glioma[26]. However, the scarcity of prospective cohort studies about the MINDDiet limits the substantiation of these claims. Further exploration is warranted regarding the protective effects of the MINDDiet against cancer, particularly concerning neurogenic tumors. Second, previous studies were conducted in specific countries and populations. For example, Lavalette et al.[20] included 41,543 participants in France, and Bodén et al.[21] included 35,393 participants in Sweden. This constrained the generalizability of the conclusions to a broader population exhibiting diverse dietary behaviors. Third, the potential mechanisms underlying the effects of a MedDiet or MINDDiet on cancer have not been fully elucidated. A prominent feature of cancer is the induced metabolic transformation, which offers a critical target for cancer therapy[27]. Dietary factors can significantly influence tumor growth by altering cell metabolism, partly through changes in nutrient access and utilization by cancer cells[28]. Changes in dietary compositions contribute to perturbations in plasma metabolite levels, subsequently affecting metabolite concentrations in the tumor microenvironment. The metabolic activity of cancer cells could be altered in response to changes in metabolite concentrations in their local environment[29]. Therefore, our hypothesis is that dietary factors may influence the levels of metabolites in the tumor microenvironment, thereby modulating cancer cell metabolism and ultimately impacting tumor growth. Metabolomics enables the comprehensive analysis of small molecules (metabolites) in biological samples, thereby offering valuable insights into the dynamic metabolic processes influenced by diet[30]. In the field of metabolomics, Nuclear Magnetic Resonance (NMR) spectroscopy has become one of the primary analytical methods due to its notable advantages such as high reproducibility and quantitative capabilities, non-selective and non-invasive nature, and the ability to identify unknown metabolites in complex mixtures[31].

In this work, we aim to investigate the associations between adherence to a MedDiet or a MINDDiet with the risks of overall and 22 specific cancers using data from the UK Biobank. In addition, we aim to identify metabolites associated with overall cancer risk from the original pool of 168 NMR metabolites and explore the mediating roles of these identified metabolites in the associations between diet adherence and overall cancer risk.

## Results
### Descriptive information

Statistics of the baseline information for the 187,485 individuals analyzed for associations between diet scores and cancer risk are shown in Table 1. Over the follow-up period, spanning from 0.1 to 15.7 years, for each participant (median = 13.2 years), a total of 26,391 cancer cases were diagnosed, with 12,437 (47.1%) occurring in females and 13,954 (52.9%) in males. Detailed statistics of participants with and without

overall cancer can also be found in Table 1. The characteristics of 22 specific cancers are summarized in Supplementary Data 1.

As shown in Table 1, the median age at recruitment for those subsequently diagnosed with cancer was 5 years older than participants who were not diagnosed with cancer. Compared to participants with no cancer, participants diagnosed with cancer were more likely to have a family history of cancer, be smokers or alcohol consumers, and have a higher body mass index (BMI) and waist-hip ratio (WHR). The median ($P_{25}$, $P_{75}$) values of the Mediterranean diet adherence screener (MEDAS) and MIND scores showed no significant differences between cancer and non-cancer groups. Statistical characteristics of the study

**Table 1 | Baseline characteristics of UK Biobank participants included in diet adherence study (N = 187,485)**

| Characteristics | Participants (N = 187,485) | Non-cancer (N = 161,094) | Overall Cancer (N = 26,391) |
|---|---|---|---|
| **Age at recruitment (year)** | 57 (49, 62) | 56 (49, 62) | 61 (56, 65) |
| **Sex** | | | |
| Female | 101079 (54.2%) | 89272 (55.4%) | 12437 (47.1%) |
| Male | 85776 (45.8%) | 71822 (44.6%) | 13954 (52.9%) |
| **Completion of full-time education** | | | |
| No | 80992 (43.2%) | 70167 (43.6%) | 10825 (41.0%) |
| Yes | 106493 (56.8%) | 90927 (56.4%) | 15566 (59.0%) |
| **Average total household income (per year)** | | | |
| < 30,999£ | 84580 (45.1%) | 71166 (44.2%) | 13414 (50.8%) |
| ≥ 30,999£ | 102905 (54.9%) | 89928 (55.8%) | 12977 (49.2%) |
| **Townsend deprivation index** | | | |
| Low | 61878 (33.0%) | 52640 (32.7%) | 9238 (35.0%) |
| Medium | 61866 (33.0%) | 52905 (32.8%) | 8961 (34.0%) |
| High | 63741 (34.0%) | 55549 (34.5%) | 8192 (31.0%) |
| **Family history of cancer** | | | |
| No | 138612 (73.9%) | 119945 (74.5%) | 18667 (70.7%) |
| Yes | 48873 (26.1%) | 41149 (25.5%) | 7724 (29.3%) |
| **Smoking status** | | | |
| Never | 106872 (57.0%) | 93290 (57.9%) | 13582 (51.5%) |
| Former occasional smoker | 22134 (11.8%) | 18922 (11.7%) | 3212 (12.2%) |
| Former regular smoker | 43679 (23.3%) | 36274 (22.5%) | 7405 (28.1%) |
| Current smoker | 14800 (7.9%) | 12608 (7.8%) | 2192 (8.3%) |
| **Alcohol drinking status** | | | |
| Never | 11695 (6.2%) | 10184 (6.3%) | 1511 (5.7%) |
| Occasional drinker | 39091 (20.9%) | 34100 (21.2%) | 4991 (18.9%) |
| Regular drinker | 136699 (72.9%) | 116810 (72.5%) | 19889 (75.4%) |
| **Physical activity (days/week)** | 2.5 (1.0, 4.0) | 2.5 (1.0, 4.0) | 2.5 (1.0, 4.0) |
| **Sleep duration (hour)** | 7 (7, 8) | 7 (7, 8) | 7 (7, 8) |
| **BMI (kg/m$^2$)** | 26.3 (23.8, 29.3) | 26.2 (23.7, 29.3) | 26.5 (24.0, 29.4) |
| **WHR, %** | 0.86 (0.80, 0.93) | 0.86 (0.79, 0.93) | 0.88 (0.81, 0.94) |
| **SBP (mmHg)** | 137 (125, 151) | 136 (124, 150) | 140 (128, 154) |
| **Dietary Energy intake, calories/d** | 2041 (1699, 2442) | 2036 (1695, 2439) | 2071 (1722, 2460) |
| **MEDAS score** | 5.0 (4.0, 6.0) | 5.0 (4.0, 6.0) | 5.0 (4.0, 6.0) |
| **MIND score** | 6.5 (5.0, 7.5) | 6.5 (5.0, 7.5) | 6.5 (5.0, 7.5) |

*BMI* body mass index, *WHR* waist-hip ratio, *SBP* systolic blood pressure, *MEDAS* Mediterranean diet adherence screener, *MIND* Mediterranean-DASH Diet Intervention for Neurodegenerative Delay. Values for categorical factors are presented as number (percentage), and for continuous variables are presented as median ($P_{25}$, $P_{75}$).

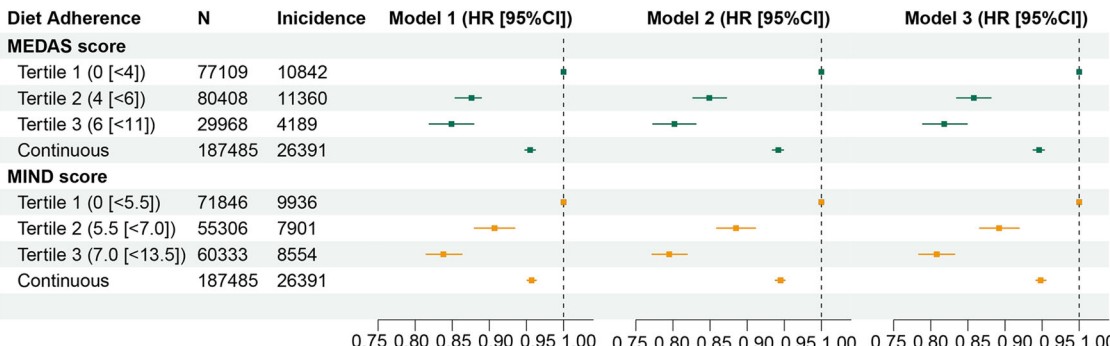

**Fig. 1 | Associations between MEDAS and MIND scores and overall cancer risk (N = 187,485).** HR hazard ratio, CI confidence interval, MEDAS Mediterranean diet adherence screener, MIND Mediterranean-DASH Diet Intervention for Neurodegenerative Delay. *HR* and 95%*CI* were estimated by Cox regression (two-sided Wald test). The green color indicates the MEDAS score, while the orange color indicates the MIND score. The gray dashed line represents the null (*HR* = 1). Each point shows the point estimate of *HR* from Cox regression. Bars show 95%CI. Model 1: adjusted for energy intake; Model 2: further adjusted for demographic factors (including sex, completion of full-time education, average total household income, Townsend deprivation index, and family history of cancer); Model 3: further adjusted for behavioral risk factors (including smoking status, alcohol drinking status, physical activity, sleep duration, BMI, WHR, and SBP). MEDAS and MIND scores were treated as continuous and categorical variables, respectively, in all models (Detailed *HR*, 95%*CI*, and *p*-value are summarized in Supplementary Data 7, source data are provided as a Source Data file).

population by tertiles of MEDAS and MIND scores are shown in Supplementary Data 2 and Supplementary Data 3, respectively. According to these two tables, participants with higher adherence to MEDAS and MIND were more likely to be female, have higher household income, have lower BMI and WHR, and be more physically active.

It should be pointed out that the number of study populations for analyses of metabolite identification and mediating roles of metabolites was different from the number of our study population. However, the statistics of participants with and without overall cancer, alongside the distribution of 22 specific types of cancer incident events, were very similar. Details are shown in Table 1 and Supplementary Data 1, 4–6.

### Associations between diet scores and cancer risk

Associations between MEDAS and MIND scores with overall cancer risk were examined using Cox regression models, with diet scores treated as categorical and continuous variables respectively. We used 3 different models, adjusting for different covariates to capture these associations. Results from the 3 different models, adjusting for different covariates, consistently showed that higher MEDAS and MIND scores were significantly associated with a lower overall cancer risk, as shown in Fig. 1. When scores were treated as categorical variables (reference = tertile 1), the *HR* (95%*CI*) values for MEDAS and MIND scores at tertile 3 using model 1, adjusting for energy intake, were 0.849 (0.819, 0.879) ($P_{trend}$ = 7.53 × 10$^{-21}$) and 0.838 (0.815, 0.863) ($P_{trend}$ = 3.25 × 10$^{-33}$) respectively; after further adjusting for demographic factors (including sex, completion of full-time education, average total household income, Townsend deprivation index, and family history of cancer) in model 2, *HR* (95%*CI*) values were 0.802 (0.773, 0.831) ($P_{trend}$ = 1.24 × 10$^{-24}$) and 0.795 (0.772, 0.819) ($P_{trend}$ = 3.22 × 10$^{-53}$) respectively; similar results were observed in model 3 with additional adjustment for behavioral risk factors (including smoking status, alcohol drinking status, physical activity, sleep duration, BMI, WHR, and systolic blood pressure [SBP]), with *HR* (95%*CI*) values of 0.818 (0.789, 0.849) ($P_{trend}$ = 2.16 × 10$^{-31}$) and 0.808 (0.784, 0.832) ($P_{trend}$ = 2.16 × 10$^{-49}$), respectively. Consistent results were observed when diet scores were treated as a continuous variable as shown in Fig. 1. Detailed *HR* values are summarized in Supplementary Data 7.

In addition, we investigated the associations between specific dietary components and overall cancer risk. The results are presented in Supplementary Fig. 1. Our findings revealed that vegetables, fruits, berries, nuts, whole grains, seafood, legumes, and wine were associated with a lower risk of overall cancer. Conversely, butter, margarine or cream, cheese, red meat, and products, fast/fried food, pastries, and sweets were associated with a higher risk of overall cancer. These beneficial foods mostly align with the dietary recommendations advocated by these two diet scores, while the risky ones are discouraged. These findings further demonstrate the protective effects of both diet patterns against overall cancer.

For 22 specific cancers, we examined the associations using model 3, with diet scores as categorical variables. As shown in Fig. 2, higher adherence to the MEDAS score demonstrated significant associations with reduced risks of 14 specific cancers, namely esophagus cancer, colorectal cancer, lung cancer, malignant melanoma, breast cancer, uterus and cervix cancer, prostate cancer, testis cancer, kidney cancer, bladder cancer, brain cancer, thyroid cancer, non-Hodgkin lymphoma, and leukemia. While, adherence to the MIND score exhibited inverse associations with 13 specific cancers, including head and neck cancer, esophagus cancer, colorectal cancer, lung cancer, malignant melanoma, breast cancer, uterus and cervix cancer, prostate cancer, kidney cancer, thyroid cancer, non-Hodgkin lymphoma, multiple myeloma, and leukemia. Significant protective effects of both MEDAS and MIND scores at tertile 2 and tertile 3 were observed in colorectal cancer, lung cancer, breast cancer, and prostate cancer. Detailed *HR* values of all 22 specific types of cancer are summarized in Supplementary Data 8.

We performed stratified analysis to examine associations between diet scores and overall cancer risk in different subgroups categorized by demographic and lifestyle variables, including sex, completion of full-time education, family history of cancer, smoking status, alcohol drinking status, and BMI.

For overall cancer, as detailed in Supplementary Fig. 2, consistently negative associations between MEDAS and MIND scores with the risk were observed across different subgroups. However, a significant interaction between MIND score with sex was observed ($P_{interaction}$ = 0.018), indicating a stronger association in males (*HR* [95%*CI*]: 0.880 [0.862, 0.897]) compared to females (*HR* [95%*CI*]: 0.918 [0.898, 0.938]).

Expanding our analysis to specific cancers, as shown in Supplementary Fig. 3, we observed consistently negative associations of MEDAS and MIND scores with most cancer risks across different subgroups. Nevertheless, significant differences in associations were evident within certain subgroups. For stomach cancer, significant interactions were observed between MEDAS and MIND scores with smoking status ($P_{interaction}$: 0.012 and 0.014 respectively); for colorectal cancer, significant interactions were observed between MEDAS and

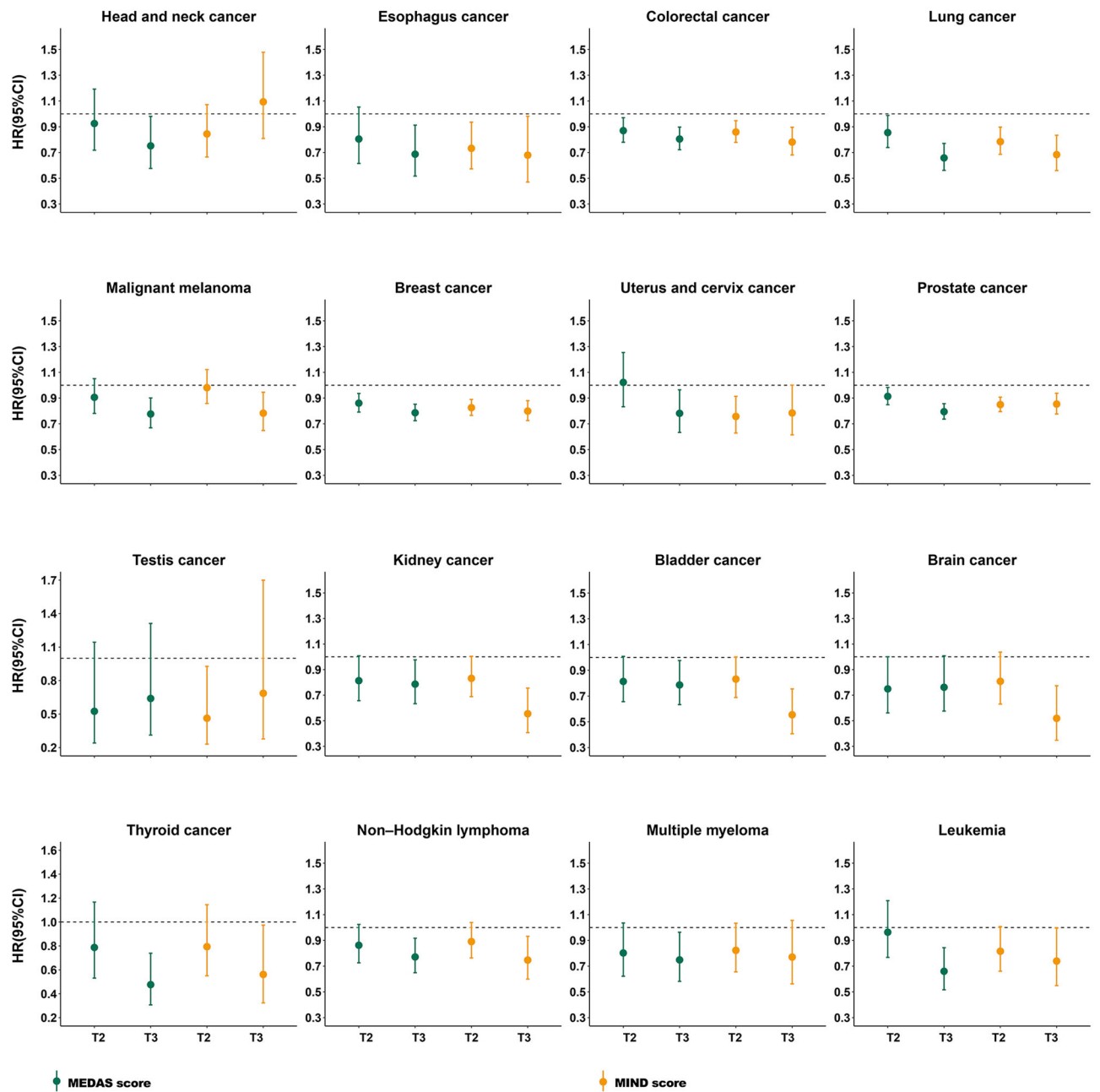

**Fig. 2 | Associations between MEDAS and MIND scores and specific types of cancer risk.** HR hazard ratio, CI confidence interval, MEDAS Mediterranean diet adherence screener, MIND Mediterranean-DASH Diet Intervention for Neurodegenerative Delay; T2 Tertile 2, T3 Tertile 3, Tertile 1 scores were used as reference values in Cox regression. The green color indicates the MEDAS score, while the orange color indicates the MIND score. The sample size of study populations for each specific type of cancer risk are as follows: $N_{head\ and\ neck\ cancer}$ = 161,440; $N_{esophagus\ cancer}$ = 161,401; $N_{colorectal\ cancer}$ = 162,960; $N_{lung\ cancer}$ = 161,094; $N_{malignant\ melanoma}$ = 162,154; $N_{breast\ cancer}$ = 161,094; $N_{uterus\ and\ cervix\ cancer}$ = 161,417; $N_{prostate\ cancer}$ = 165,267; $N_{testis\ cancer}$ = 161,137; $N_{kidney\ cancer}$ = 161,596; $N_{bladder\ cancer}$ = 161,466; $N_{brain\ cancer}$ = 161,382; $N_{thyroid\ cancer}$ = 161,230; $N_{non-Hodgkinlymphoma}$ = 161,879; $N_{multiple\ myeloma}$ = 161,457; $N_{leukemia}$ = 161,513. The *HR* and *95%CI* were estimated using Cox regression with adjustment for covariates in model 3 (two-sided Wald test). Each point shows the point estimate of *HR* from Cox regression. Bars show 95%CI (Detailed *HR*, *95%CI*, and *p*-value are summarized in Supplementary Data 8, source data are provided as a Source Data file).

MIND score with sex ($P_{interaction}$: 0.013 and 0.006 respectively) and smoking status($P_{interaction}$: < 0.001 and 0.007 respectively); for liver cancer, significant interaction between MEDAS score with BMI ($P_{interaction}$ = 0.009) was observed; for pancreas cancer, significant interaction between MIND score with sex ($P_{interaction}$ = 0.019) was observed; for lung cancer, significant interaction between MIND score with smoking status ($P_{interaction}$ = 0.001) was observed; for Hodgkin lymphoma, significant interaction between MEDAS score with completion of full-time education ($P_{interaction}$ = 0.024) was observed; for

leukemia, significant interaction between MIND score with completion of full-time education ($P_{interaction}$ = 0.046) was observed. Detailed *HR* values within subgroups categorized by the relevant variables and $P_{interaction}$ values are summarized in Supplementary Data 9.

**Identification of metabolites associated with overall cancer risk**
We used a sequential analysis strategy that combined the Cox regression model, elastic net model (ENM), and gradient boost model (GBM) to select metabolites associated with overall cancer from the

**Table 2 | Associations between the top ten identified metabolites and overall cancer risk**

| Metabolites | Group | HR | 95%CI | P | FDR |
|---|---|---|---|---|---|
| Total Lipids in VLDL | Total lipids | 0.972 | 0.953–0.991 | 0.005 | 0.006 |
| Total Cholines | Other lipids | 0.921 | 0.904–0.939 | $7.19 \times 10^{-18}$ | $1.80 \times 10^{-17}$ |
| Omega-3 Fatty Acids | Fatty acids | 0.880 | 0.864–0.897 | $4.18 \times 10^{-41}$ | $4.19 \times 10^{-40}$ |
| Tyrosine | Acids | 0.935 | 0.918–0.952 | $3.75 \times 10^{-13}$ | $6.25 \times 10^{-13}$ |
| Glucose | Glycolysis related metabolites | 0.955 | 0.938–0.972 | $2.68 \times 10^{-7}$ | $3.84 \times 10^{-7}$ |
| Citrate | Glycolysis related metabolites | 0.902 | 0.885–0.918 | $9.64 \times 10^{-29}$ | $4.83 \times 10^{-28}$ |
| Creatinine | Fluid balance | 0.976 | 0.957–0.995 | 0.012 | 0.012 |
| Albumin | Fluid balance | 1.051 | 1.029–1.073 | $1.99 \times 10^{-6}$ | $2.49 \times 10^{-6}$ |
| Free Cholesterol in IDL | Lipoprotein subclasses | 0.907 | 0.890–0.924 | $6.10 \times 10^{-25}$ | $2.03 \times 10^{-24}$ |
| Total Lipids in Large HDL | Lipoprotein subclasses | 0.908 | 0.888–0.928 | $1.29 \times 10^{-17}$ | $2.60 \times 10^{-17}$ |

*VLDL* very low-density lipoprotein, *IDL* intermediate-density lipoprotein, *HDL* high-density lipoprotein, *HR* hazard ratio, *CI* confidence interval, *FDR* false discovery rate. *HR* and *95%CI* were estimated by Cox regression in the testing set ($N = 80,921$), $HR > 1$ (or $HR < 1$) indicates metabolites had a positive (or negative) association with overall cancer risk. *FDR* is an indicator used to adjust the *p*-value and to control for false detective rate in multiple hypothesis testing. Associations were considered significant when $FDR < 0.05$ (two-sided Wald test).

original pool of 168 NMR metabolites in a population of 202,303 individuals. Cox regression showed that 114 of the 168 baseline metabolites had independent associations with overall cancer, adjusting for covariates as described in the method section. We then performed ENM analyses to narrow down the list of candidate metabolites. Among the 114 metabolites, 42, 38, 22, and 35 metabolites were identified with α values of 0.25, 0.50, 0.75, and 1.00, respectively (Supplementary Fig. 4a–d and Supplementary Data 10). Twenty-one shared metabolites emerged from the above ENM analysis with different α values (Supplementary Fig. 4e). Subsequently, the relative importance of the 21 shared metabolites was evaluated using GBM. A more detailed description of the method used to identify metabolites is given in the method section.

Finally, we identified the top 10 metabolites associated with overall cancer risk, as shown in Supplementary Fig. 4f, namely total lipids in VLDL, total cholines, omega-3 fatty acids, tyrosine, glucose, citrate, creatinine, albumin, free cholesterol in IDL, and total lipids in large HDL. These metabolites belong to different categories (e.g., total lipids, other lipids, fatty acids, acids, glycolysis-related metabolites, fluid balance, and lipoprotein subclasses) and are involved in various metabolic pathways.

Table 2 shows the associations between the 10 identified metabolites and overall cancer risk analyzed using a Cox regression model. We observed negative associations between these metabolites and overall cancer risk, with the exception of albumin, which showed a positive association.

Similarly, as shown in Fig. 3, the majority of identified metabolites were negatively associated with risks of multiple types of cancer. For example, omega-3 fatty acids, citrate, and free cholesterol in IDL were significantly inversely associated with colorectal cancer, lung cancer, breast cancer, thyroid cancer, non-Hodgkin lymphoma, and leukemia. Albumin, which was identified as a risk factor for overall cancer, pancreas cancer, malignant melanoma, prostate cancer, and leukemia, was found to be negatively associated with specific cancer types such as liver cancer, lung cancer, bladder cancer, and multiple myeloma. *HR* values for the associations between 10 metabolites and risks of 22 specific cancers are listed in Supplementary Data 11.

### Mediating roles of metabolites
We performed mediation analysis to explore the individual and joint mediating effects of the identified metabolites on the associations between MEDAS and MIND scores and overall cancer risk in a population of 85,669 individuals with accessible information on both diet assessment and metabolites.

We found significant associations between MEDAS and MIND scores, identified metabolites, and overall cancer risk according to Supplementary Fig. 5. The mediation analysis results are given in

Supplementary Data 12. Six metabolites exhibited significant, natural, indirect effects in the association between MEDAS score and overall cancer risk. Specifically, total cholines, omega-3 fatty acids, free cholesterol in IDL, and total lipids in large HDL contributed 7.8%, 19.7%, 6.1%, and 14.1% respectively to the total effect of the MEDAS score, while total lipids in VLDL and albumin masked the associations by 4.5% and 1.9% respectively. Eight metabolites showed significant, natural, indirect effects in the association between MIND scores and overall cancer risk, where total cholines, omega-3 fatty acids, glucose, citrate, free cholesterol in IDL and total lipids in large HDL mediated 7.1%, 17.1%, 1.3%, 1.3%, 5.8% and 11.5% of the total effect respectively, while total lipids in VLDL and albumin masked 2.6% and 1.3% of the effect respectively.

Furthermore, we explored the joint mediating effect of the 10 identified metabolites. As shown in Fig. 4a, the joint mediating effect of the metabolites contributed 19.7% to the total effect of the MEDAS score on overall cancer risk, among which, total lipids in VLDL, total cholines, omega-3 fatty acids, albumin, free cholesterol in IDL, and total lipids in large HDL accounted for −9.7%, −9.7%, 18.2%, −7.8%, 6.7%, and 16.6% respectively. The joint mediating effect contributed 20.8% to the total effect of the MIND score, among which, total lipids in VLDL, total cholines, omega-3 fatty acids, citrate, albumin, free cholesterol in IDL, and total lipids in large HDL accounted for −5.3%, −7.2%, 16.0%, 3.2%, −6.2%, 6.2%, and 13.0% respectively (Fig. 4b).

### Sensitivity analyses
We performed sensitivity analyses to analyze the robustness of our findings (Supplementary Data 13–23). The conclusions remained consistent when we recalculated the MEDAS and MIND scores using different dietary assessments other than the averages; employed an alternative scoring method to obtain an alternative score to describe MedDiet adherence; excluded each of the food components in MEDAS and MIND to derive new MEDAS and MIND scores; excluded each type of cancer from the overall cancer definition; additionally adjusted for smoking intensity and alcohol intake frequency, or additionally adjusted for major nutrients including protein, fat, carbohydrate, and dietary fiber, or using different exclusion criteria. Details are provided in the method section. All these sensitivity analysis results indicated the robustness of our findings.

### Discussion
In this prospective cohort study based on a large-scale population from the UK Biobank, we have demonstrated the associations between higher MEDAS and MIND scores and a reduced risk of overall cancer. Moreover, these associations exhibited robustness and consistency across 14 and 13 specific cancers for MEDAS and MIND scores, respectively. In addition, we identified 10 metabolites

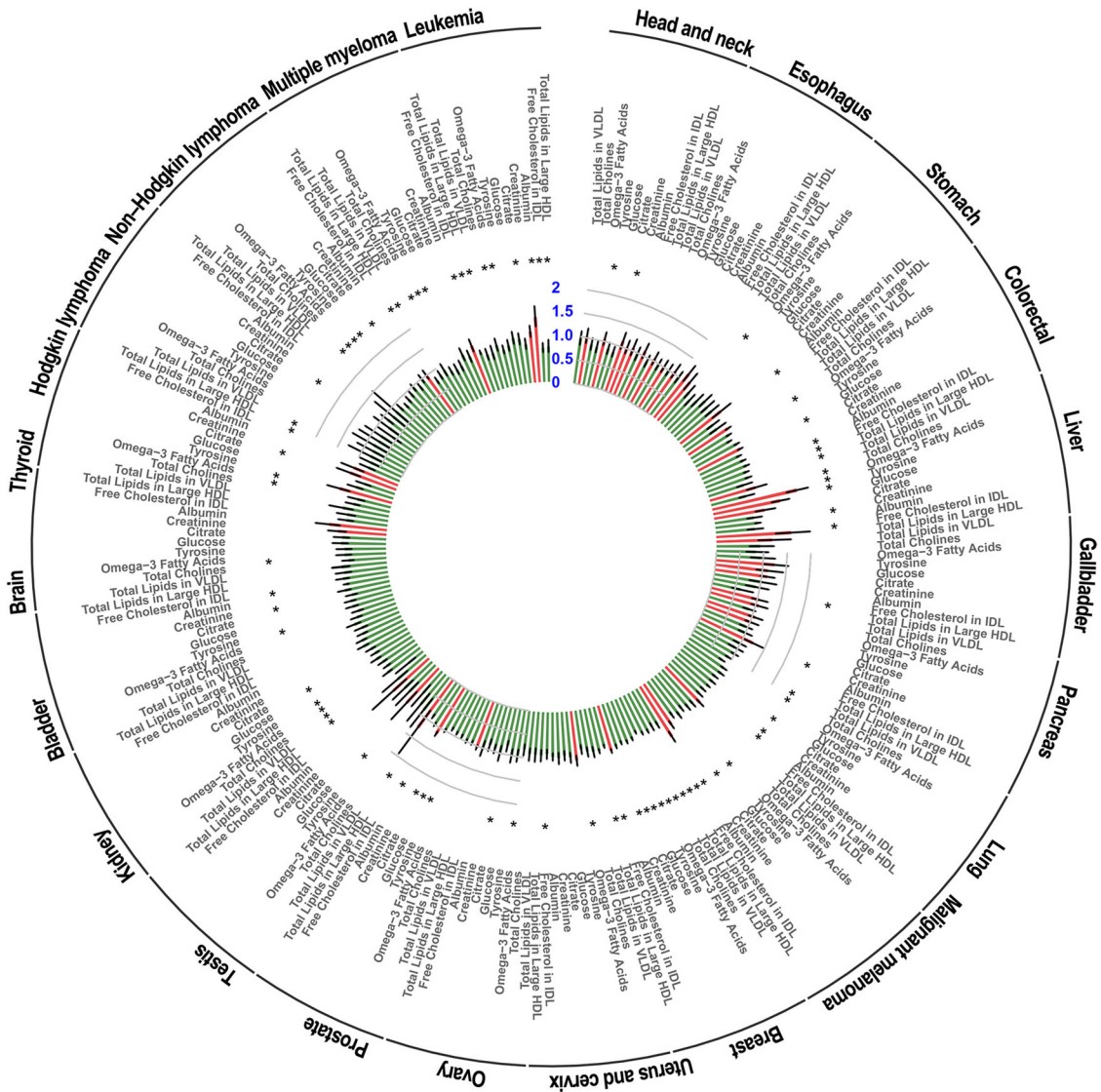

**Fig. 3 | Associations between the top 10 identified metabolites and 22 specific types of cancer risk.** HR hazard ratio, CI confidence interval. The green color represents $HR < 1$; the red color represents $HR > 1$. The black color represents the 95%CI. The blue number represents the $HR$ values. The sample size of study populations for each specific type of cancer risk are as follows: $N_{head\ and\ neck\ cancer} = 171,885$; $N_{esophagus\ cancer} = 171,877$; $N_{stomach\ cancer} = 171,795$; $N_{colorectal\ cancer} = 172,651$; $N_{liver\ cancer} = 171,787$; $N_{gallbaldder\ cancer} = 171,747$; $N_{pancreas\ cancer} = 171,859$; $N_{lung\ cancer} = 172,285$; $N_{malignant\ melanoma} = 172,202$; $N_{breast\ cancer} = 173,447$; $N_{uterus\ and\ cervix\ cancer} = 171,986$; $N_{ovary\ cancer} = 171,874$;

$N_{prostate\ cancer} = 173,401$; $N_{testis\ cancer} = 171,717$; $N_{kidney\ cancer} = 171,923$; $N_{bladder\ cancer} = 171,874$; $N_{brain\ cancer} = 171,813$; $N_{thyroid\ cancer} = 171,763$; $N_{hodgkin\ lymphoma} = 171,711$; $N_{non-Hodgkin\ lymphoma} = 172,085$; $N_{multiple\ myeloma} = 171,870$; $N_{leukemia} = 171,908$. The $HR$ and 95%CI were estimated using Cox regression with adjustment for covariates except for dietary energy intake in model 3 (two-sided Wald test). * $HR$ values with significant $p$ ($p < 0.05$) based on Cox regression. Detailed $HR$, 95%CI, and p-value are summarized in Supplementary Data 11, source data are provided as a Source Data file.

associated with overall cancer, most of which robustly exhibited significant associations across multiple types of cancers. Our mediation analysis results further indicated that several of the identified metabolites (e.g., total cholines, omega-3 fatty acids, free cholesterol in IDL, and total lipids in large HDL) played a mediating role in the effects of diet score on overall cancer risk, whereas total lipids in VLDL and albumin masked the effects. Our results contribute to a more comprehensive understanding of the preventative potential of specific dietary patterns, and the mediating roles of metabolites in these preventive effects.

Generally, we observed significant negative associations between both MEDAS and MIND scores and the risk of overall, and several specific cancers, indicating that both dietary patterns, characterized by abundant consumption of vegetables, legumes, nuts, grains, fish, and seafood, while reducing the consumption of butter, margarine,

cream, cheese, red meat and products, as well as pastries and sweets, provide protective effects against cancer.

The significant decrease in the overall cancer risk with higher adherence to MEDAS we found in our study, is consistent with most reported meta-analyses[9,13,32]. We found significant protective effects of a higher MEDAS score against kidney, brain, and thyroid cancers, which has not been previously reported. The associations between a MedDiet and the risks of esophageal cancer[33], colorectal cancer[34], lung cancer[21], breast cancer[35], prostate cancer[36], and non-Hodgkin lymphoma[37] were in line with the results in most previous studies. This protective effect can be attributed to various dietary components present in the MedDiet, such as vegetables, fruit, polyphenols, fibers, phytosterols, monounsaturated, and polyunsaturated fatty acids, which have been postulated to contribute to cancer prevention[38–40]. However, a few studies have reported results that are inconsistent with

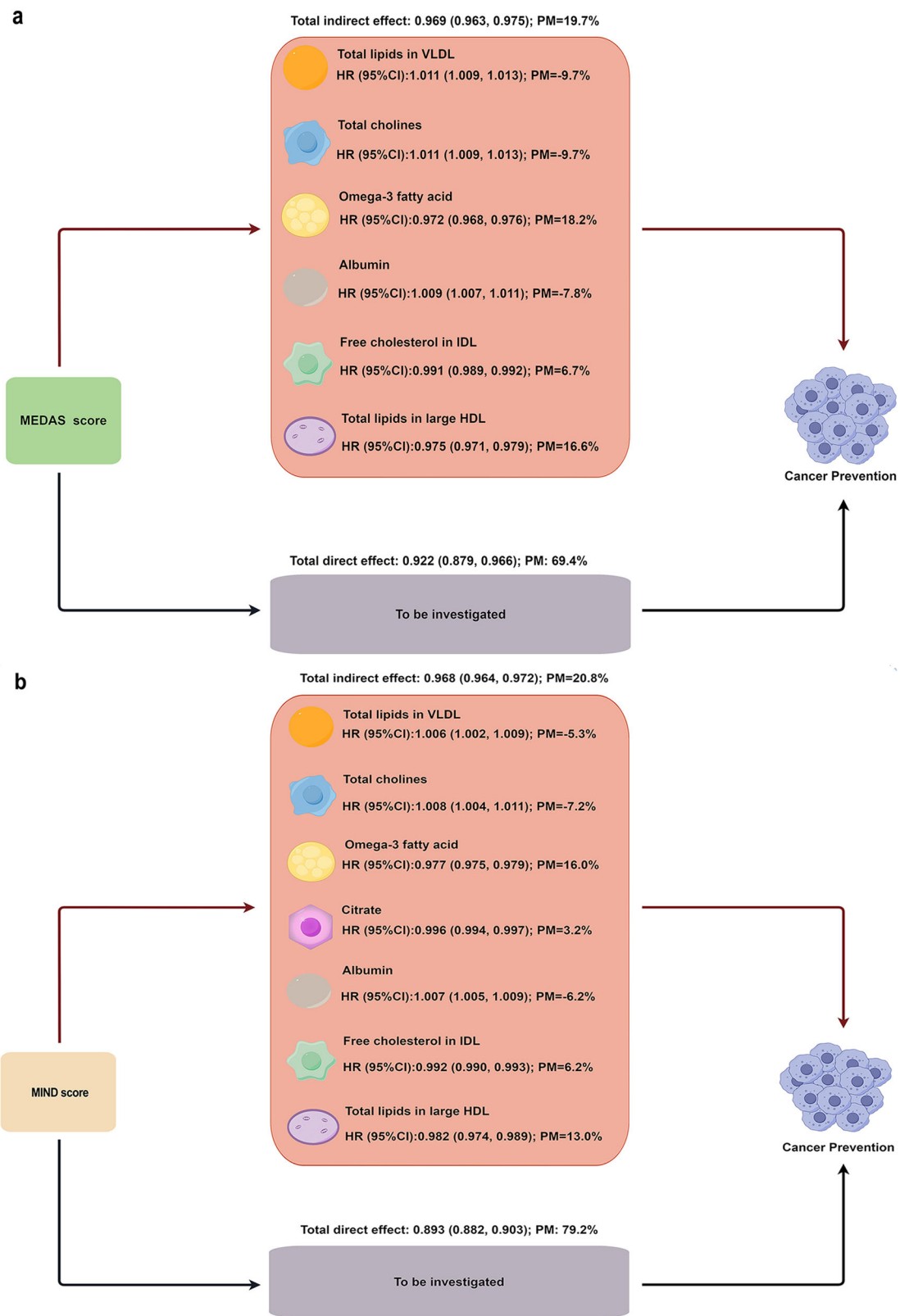

**Fig. 4 | Multiple mediation analysis with metabolites as potential mediators for the associations between MEDAS and MIND scores and overall cancer risk (*N* = 85,669).** VLDL very low-density lipoprotein, IDL intermediate-density lipoprotein, HDL high-density lipoprotein, HR hazard ratio, CI confidence interval, PM proportion mediated, MEDAS Mediterranean diet adherence screener, MIND Mediterranean-DASH Diet Intervention for Neurodegenerative Delay. **a** Multiple mediation analysis with metabolites as potential mediators for the associations between MEDAS scores and overall cancer. **b** Multiple mediation analysis with metabolites as potential mediators for the associations between MIND scores and overall cancer risk. The *HR* and 95%*CI* were estimated using Cox regression with adjustment for covariates in model 3 (two-sided Wald test). These figures are shown in Figdraw. Source data are provided as a Source Data file.

ours. For example, Lavalette et al.'s and Bodén et al.'s studies reported no significant associations between a MedDiet with overall cancer[20,21]; Dela et al.[41], Jalilpiran et al.[42] and Salvatore et al.[43] reported null associations between a MedDiet with breast cancer, prostate cancer, head cancer, and neck cancer, respectively. The discrepancies may be due to differences in study design, participant demographics, or differences in statistical methodologies. Nonetheless, the robustness and validity of our results were demonstrated by our large study population and a series of sensitivity analyses.

Our results also indicated significantly negative associations between a MINDDiet with the risk of overall cancer and 13 specific cancers, including head and neck cancer, esophagus cancer, colorectal cancer, lung cancer, malignant melanoma, breast cancer, uterus and cervix cancer, prostate cancer, kidney cancer, thyroid cancer, non-Hodgkin lymphoma, multiple myeloma, and leukemia. However, we found no significant association between a MINDDiet and brain cancer, contrary to our expectations, as the MINDDiet is derived from the MedDiet with specific modifications based on evidence from neuro-degenerative diseases[15]. Although results from a case-control study reported inverse associations of a MINDDiet with the risk of glioma[26], evidence from our study indicated no significant protective effects of a MINDDiet on neurogenic neoplasms. Compared to the MINDDiet, a MedDiet exhibited a stronger protective effect against several specific cancer types, including kidney cancer, brain cancer, and thyroid cancer. This was mainly due to the differences in the composition of the two diets: MINDDiet explicitly specifies the consumption of green leafy vegetables and berries, while MedDiet also includes other types of fruits and emphasizes moderate consumption of dairy products. Another potential explanation lies in the variations between the two dietary scoring systems in terms of the specified quantities for various components.

We identified 10 metabolites associated with overall cancer risk. Of these metabolites, total lipids in VLDL, total cholines, omega-3 fatty acids, tyrosine, glucose, citrate, creatinine, free cholesterol in IDL, and total lipids in large HDL had significant negative associations with overall cancer risk, with albumin being the only exception. Remarkably, our study underscores the clinical significance of tyrosine and albumin, both recognized for their diagnostic power across a spectrum of diseases, including overall cancer in the UK biobank[44]. Tyrosine, in particular, has garnered substantial interest in the clinical realm, owing to its intricate involvement in disease pathogenesis, notably in cancer contexts[45]. Albumin, previously established for its significant diagnostic contribution across various diseases[44] and incorporated into standard care practices[46], demonstrated a positive correlation with the risk of overall cancer in our study. Furthermore, our study reaffirms the role of total choline, omega-3 fatty acids, glucose, and citrate as endogenous biomarkers with discernible associations with overall cancer risk[45,47,48]. These metabolic signatures not only offer promise as potential diagnostic assays in primary care but also hint at shared metabolic pathways underlying carcinogenesis. Delving deeper into cancer-specific attribution profiles, our study unravels intriguing associations between metabolites and specific cancer types, shedding light on potential mechanisms of specific cancers. Omega-3 fatty acids, renowned for their protective effects against gastrointestinal cancers[49], exhibit significant negative associations with diverse cancers including stomach, colorectal, liver, lung, and bladder cancers, underscoring their broader implications in cancer prevention. Moreover, our findings highlight the inverse relationships between citrate and colorectal cancer[50], ovary cancer[51], and prostate cancer[52], reaffirming its potential as a discriminative biomarker in these cancers. Conversely, elevated glucose levels emerge as a distinct vulnerability factor for liver cancer, echoing prior research[48] and emphasizing the multifaceted interplay between metabolic dysregulation and cancer susceptibility. These findings pave the way for the development of targeted therapeutic interventions and precision medicine strategies in specific cancers.

A highlight of our study is the exploration of the mediating role of metabolites in the associations between dietary patterns and overall cancer risk. The joint mediating effects of the identified metabolites contributed 19.7% and 20.8% to the total effects of MEDAS and MIND scores on overall cancer risk, respectively. Specifically, omega-3 fatty acids had the largest mediating effect among these metabolites. Both dietary patterns include foods rich in omega-3 fatty acids, such as fish and nuts. Prior studies have also recognized omega-3 fatty acids as crucial bioactive compounds that contribute to the health benefits associated with a MedDiet[53]. The potential mechanisms by which omega-3 fatty acids exert a protective effect against cancer include: reducing proinflammatory lipid derivatives, inhibiting nuclear factor-κB-induced cytokine production, and decreasing growth factor receptor signaling due to alteration in cell membrane lipid rafts[54]. We also found that total choline levels exert a negative mediating effect, which indicates that adherence to a MedDiet or a MINDDiet may reduce overall cancer risk by downregulating choline levels. Prior studies have illustrated that the absorption and metabolization of choline are influenced by components of Med- and MIND-Diets, such as eggs, fish, grains, meat, milk, soybeans, and potatoes[55]. Previous studies have also suggested that choline has multiple roles in the development of cancer, including influencing DNA methylation, disrupting DNA repair, modifying cell signaling mediated by intermediary phospholipid metabolites, and supporting cell membrane synthesis to promote cell proliferation[56]. The choline metabolite profile, characterized by its total choline-containing compounds, is increasingly being recognized as a valuable adjunct for diagnosing primary malignant tumors in various organs, including the ovary[57], prostate[58], brain[59], and breast[60], and has also previously been identified as an endogenous biomarker for overall cancer[47]. In addition, total lipids in VLDL, free cholesterol in IDL, and total lipids in large HDL were identified as mediators in the associations between MEDAS or MIND scores and in overall cancer prevention, findings that are supported by previous studies. Previous studies have illustrated that lipid metabolism reprogramming is a novel hallmark of malignant tumors[61]. The bioactive compounds present in the ingredients of a MedDiet or MINDDiet have the potential to modulate lipid metabolism levels[53]. It has also been reported that cancer cell proliferation relies on the availability of cholesterol[62], the levels of which are regulated by these bioactive compounds[5]. Therefore, our hypothesis is supported by the logical inference that the metabolic changes we observed in this study were to some extent, influenced by long-term dietary patterns, thereby modulating tumor cell metabolism and ultimately impacting cell function.

Our study has several notable strengths. First, we demonstrate consistent and significant associations between higher MEDAS or MIND scores with reduced overall cancer risk. Second, to the best of our knowledge, this study stands out in identifying specific metabolites, such as total cholines, omega-3 fatty acids, free cholesterol in IDL, and total lipids in large HDL, as potential mediators behind the effects of following a MedDiet or MINDDiet on overall cancer risk. Third, a substantially large study population ensured the statistical power of our model and supported the reliability of our findings. Furthermore, we conducted a series of sensitivity analyses, affirming the robustness and validity of our results.

However, several limitations should also be acknowledged. (1) Causal inference is beyond the scope of current research, and concerns about reverse causation exist regarding diet patterns and cancer, as specific cancers may influence dietary changes. (2) There is a significant time gap between the last update for dietary and cancer incidence data. The dietary data was last updated in 2012; however, the last follow-up for the incidence of cancer was conducted up until November 31, 2022, in England, August 31, 2022, in Scotland, and May 31, 2022 in Wales. This temporal misalignment could introduce some

bias, as participants may have altered their dietary behavior patterns during this time period. (3) We used an average diet score, which can better capture participants real dietary habits in our model. However, 75,277 (40.2%) participants only had a single assessment as their baseline, potentially deviating from their true dietary patterns and introducing some bias into our results. Nevertheless, sensitivity analysis, including participants with at least two dietary assessments, showed consistency with our main model, indicating the robustness of our findings. However, it is crucial to acknowledge that a limitation still exists due to the potential for bias in self-reported dietary information. (4) The lack of information about olive oil or sofrito (a traditional Mediterranean sauce) in the dietary assessment may introduce bias for the evaluation of MEDAS and MIND scores. However, previous studies have shown that only a small proportion of people in the UK have reached the threshold of achieving a full score for olive oil[63], suggesting that the lack of olive oil intake information is likely minimal. In addition, we conducted sensitivity analyses excluding specific diet components individually from the MEDAS and MIND scores and obtained robustness results.

In conclusion, our study presents compelling evidence supporting the associations between increased adherence to a MedDiet or MINDDiet with significant reductions in overall and several specific cancers. Moreover, we identified 10 metabolites that exhibited significant associations with the risk of overall cancer, most of which are related to multiple types of cancer, which demonstrates the shared metabolic pathways implicated in carcinogenesis. Our study further reveals the individual and joint mediating roles of the identified metabolites in the associations between MEDAS and MIND scores with overall cancer risk. These findings contribute to deepening our understanding of the intricate connections between dietary patterns, metabolites, and cancer development, and may also hold implications for the development of targeted interventions for cancer prevention and management.

## Methods

### Ethics statement
Prior to enrollment, all participants provided written informed consent in accordance with the guidelines outlined in the Helsinki Declaration. This study was reviewed and approved by the NHS National Research Ethics Service (Ref: 11/NW/0382).

### Study population and design
Our study population is based on the UK Biobank, a resource library collecting and storing biological samples of participants, as well as physiological, pathological, socio-economic, and other information[64]. The UK Biobank study recruited over 500,000 participants aged 37–73 years between 2006 and 2010 from 22 assessment centers throughout England, Scotland, and Wales. Follow-ups to record incidences of cancer were conducted up until November 31, 2022, in England, August 31, 2022, in Scotland, and May 31, 2022, in Wales.

First, we investigated the associations between diet scores and risk of overall, and 22 specific cancers. After excluding participants who had a pre-existing cancer diagnosis, those lost to follow-up, individuals with missing dietary assessment information before cancer onset, or those with missing information for more than 80% covariates, we were left with a study population of 187,485 participants. Second, we identified metabolites associated with overall cancer, from the original pool of 168 metabolites. For this analysis, we excluded participants who had a pre-existing cancer diagnosis, were lost to follow-up, used lipid-lowering drugs before blood collection, and those with missing data on baseline metabolites. In addition, we excluded participants lacking information on more than 80% of the basic confounding factors. The total number of participants included was 202,303. Finally, we explored the mediating roles of the identified metabolites in how diet affects overall cancer risk. This final analysis involved 85,669 participants, constituting a shared subset between the two aforementioned studies, who had both diet assessment and metabolite data. The flowchart of our study design is shown in Supplementary Fig. 6. It should be pointed out that though the numbers of study populations varied for each of the analyses due to the differing missing variables among participants, the statistical characteristics of the populations in the 3 analyses were very similar, as can be seen from Table 1 and Supplementary Data 1, 4–6.

Our study adhered to the STROBE (Strengthening the Reporting of Observational Studies in Epidemiology) guidelines to ensure transparent and comprehensive reporting of observational studies. In addition, we followed the AGREMA (A Guideline for Reporting Mediation Analyses) guidelines to accurately report the mediation analysis conducted in this research.

### Health endpoint
The health endpoint for this study was cancer onset, which was confirmed by individual record links to national cancer, hospital episode statistics, or death records. We investigated overall cancer, and 22 specific cancers, including head and neck cancer, esophagus cancer, stomach cancer, colorectal cancer, liver cancer, gallbladder cancer, pancreas cancer, lung cancer, malignant melanoma, breast cancer, uterus and cervix cancer, ovary cancer, prostate cancer, testis cancer, kidney cancer, bladder cancer, brain cancer, thyroid cancer, Hodgkin lymphoma, non-Hodgkin lymphoma, multiple myeloma, and leukemia. Overall and specific cancers were defined according to the International Classification of Diseases (ICD)-10 codes as shown in Supplementary Data 24 in the Supplementary Materials.

### Dietary assessment
Dietary information was collected using the Oxford WebQ, which is a validated web-based, 24-hour self-administered dietary assessment tool for large observational studies[65]. Previous large-scale cohort studies have demonstrated that 24-hour dietary assessment ensured a valid estimation of usual dietary intake[66]. During each assessment, participants are required to record the number of standard portions for 206 foods and 32 beverages consumed during the previous 24-hour period. Participants had a total of up to 5 dietary assessments, including the baseline assessment between April 2009 and September 2010 and a follow-up assessment every 3 to 4 months between February 2011 and June 2012. In order to reduce within-subject variation[67], we calculated the dietary score for each dietary assessment and then calculated the average dietary score of all assessments for each participant. The scoring method is described in the following section.

### MEDAS score
We used the MEDAS scoring method to quantify adherence to the MedDiet based on the dietary assessment recorded by Oxford WebQ. The MEDAS scoring method originated from the Prevención con Dieta Mediterránea (PREDIMED) trial and has been widely used in both trials and observational studies[68,69]. MEDAS has been validated in a prospective study by Gregory et al. conducted in the UK[70], has received endorsement from the American Heart Association, and has become a rapid dietary assessment screening tool in clinical practice, due to its reliability and practical utility[71]. The MEDAS screener was calculated with a binary assessment for habitual intake of the 14 food components (Supplementary Data 25). Participants receive 1 point if their consumption meets a pre-defined cut-off of each food component, as shown in Supplementary Data 25, if not, they receive 0 points. We then derived the MEDAS score for each participant by summing up the scores obtained for each of the 14 food components assessed. The maximum MEDAS score in this study was 12, as the information on olive oil and sofrito use was not available in the UK Biobank data.

## MIND score

We utilized the MIND score methods to quantify adherence to the MIND Diet. The calculation of the MIND score incorporated 15 food components, with 10 favorable food groups and 5 non-favorable food groups, as described in Supplementary Data 26. Regarding the MINDDiet score components, we aggregated the consumption of each portion associated with the food component and subsequently assigned a concordance score of 0, 0.5, or 1. Higher MIND scores indicate greater adherence to the MINDDiet[18]. As olive oil consumption was not available in UK Biobank data, the maximum MIND score in our study was 14.

## Metabolite quantification

A high-throughput targeted NMR-metabolomics platform was used to perform metabolomic profiling on baseline EDTA plasma samples from a randomly selected subset of ~280,000 UK Biobank participants. Each sample contained the original 168 metabolites in absolute levels (mmol/L), including categories such as fatty acids, glycolysis metabolites, ketone bodies, amino acids, lipids, and lipoproteins. Detailed protocols for sample collection and metabolomic quantification have been described in previous studies[72].

## Covariates

According to previous studies[25,45,73], the covariates selected include age at recruitment (continuous variable), sex (binary variable: female, male; defined by biological attribute and self-reported), completion of full-time education (binary variable: no, yes), average total household income (binary variable: < 30,999£, ≥ 30,999£), Townsend deprivation index (categorical variable: low, medium, high), family history of cancer (binary variable: no, yes), smoking status (categorical variable: never, former occasional smoker, former regular smoker, current smoker), alcohol drinking status (categorical variable: never, occasional drinker, regular drinker; categorized as regular if more than once a week, and as an occasional drinker if one to three times a month or only on special occasions), physical activity (continuous variable), sleep duration (continuous variable), BMI (continuous variable), WHR (continuous variable), SBP (continuous variable), and dietary energy intake (continuous variable). The dietary energy intake information was obtained from the Oxford WebQ, and information of the other covariates was obtained via a touch-screen questionnaire and by physical measurements. Further details can be found in Supplementary Data 27.

## Statistical analyses

Multiple imputations were used to fill in missing data when the missing rate of covariates for the individual was below 20%. It was assumed that the missing values of 168 metabolites were attributable to the limit of detection; and therefore, these missing values were imputed using half of the minimum detectable value. The individual metabolite concentrations were then natural logarithmic transformation (ln[x + 1]) and standardized to z-scores.

Cox regression analysis was used to estimate the associations between MEDAS and MIND scores with overall, and 22 specific cancer risks, taking recruitment age as the timescale[73]. MEDAS and MIND scores were treated as continuous or categorical variables using the lowest tertile as reference, in separate models. We employed 3 models: model 1 with dietary energy intake adjusted; model 2 with further adjustment for demographic characteristics, including sex, completion of full-time education, average total household income, Townsend deprivation index, and family history of cancer; model 3 with additional adjustment for behavioral risk factors in model 2, including smoking status, alcohol drinking status, physical activity, sleep duration, BMI, WHR, and SBP. The covariates in model 3 were employed for all the subsequent analyses, except for the identification of metabolites, for which dietary energy intake was not adjusted.

Results were expressed as hazard ratio (HR) and 95% confidence interval (CI).

Likelihood ratio tests were used to assess the heterogeneity of the associations between MEDAS and MIND scores with overall and specific cancer risks in different groups stratified by sex (female, male), completion of full-time education (no, yes), family history of cancer (no, yes), smoking status (no, yes), alcohol drinking status (no, yes), and BMI ( < 25 kg/m², ≥ 25 kg/m²).

We used a sequential analysis strategy that combined the Cox regression model, ENM, and GBM to select metabolites associated with overall cancer risk from the original pool of 168 NMR metabolites, in 202,303 individuals. First, all participants (n = 202,303) were randomly assigned to a training (n = 121,382) or a testing set (n = 80,921). In the training set, a Cox regression model was used to test for individual associations between metabolites and overall cancer risk (Results are shown in Supplementary Data 28). FDR (false discovery rate) adjusted P-values were calculated using the Benjamini-Hochberg method. Then, ENM was used to select metabolites from an extensive pool of candidates following the Cox regression analysis, taking into consideration potential multicollinearity between metabolites. ENM is a supervised machine learning algorithm that seamlessly integrates the least absolute shrinkage and selection operator (LASSO) regression and ridge regression. It is known for its excellent probabilistic interpretation of variables and suitability for disease prediction[74]. A 10-fold cross-validation was used to select optimal $\lambda$ and $\beta$ coefficients for the ENM models. To achieve sparsity of the model, the crossover and over-lapped metabolites were identified with α values of 0.25, 0.50, 0.75, and 1.00 from the ENM models. Subsequently, we used GBM analysis to evaluate the relative importance of metabolites. Finally, the top 10 metabolites from the GBM analysis were considered as identified metabolites. The associations between these metabolites with overall and 22 specific types of cancer risks were tested by the Cox regression model in the testing set.

Mediation analyses were used to determine the potential mediating effects of the identified metabolites on the associations between MEDAS and MIND scores with overall cancer risk (R package "mma"). In addition to the individual mediating effects of metabolites, we also assessed the joint mediating effect of the identified metabolites. The mediating effect was expressed as proportion mediated (PM), using the following equation:

$$PM = \frac{HR^{NDE} \times (HR^{NIE} - 1)}{HR^{NDE} \times HR^{NIE} - 1} \tag{1}$$

where, $HR^{NDE}$ denoted HR of natural direct effects and $HR^{NIE}$ was HR of natural indirect effects.

## Sensitivity analyses

Sensitivity analyses were conducted to assess the robustness of our study: (1) we computed the MEDAS and MIND score based solely on the baseline dietary assessment instead of average dietary intake; (2) we only included participants with at least two dietary assessments before cancer onset; (3) we employed a different scoring method to obtain an alternative score to describe MedDiet adherence[75] and examined the association between the new score and overall cancer risk; (4) we excluded each of the food components in the dietary patterns to derive new MEDAS and MIND scores; (5) we excluded each type of specific cancer from the definition of overall cancer; (6) we additionally adjusted for smoking intensity (pack years of smoking) and alcohol intake frequency in our main model; (7) we additionally adjusted for major nutrients (protein, fat, carbohydrate, and dietary fiber) in our main model; (8) we excluded participants who suffered overall cancer incidence within 2 years after completing their last dietary assessment; (9) we excluded participants with indeterminate or suboptimal overall health assessments (coding of overall health rating was greater than 2

or less than 0) to rule out potential reverse causality; (10) we excluded participants with benign tumors at recruitment.

Most of the above statistical analyses were performed using R version 4.1.1 software, and the average diet score was computed using Python version 3.11. A 2-tailed $P < 0.05$ was considered statistically significant.

## Statistics & reproducibility

No statistical method was used to predetermine the sample size. The study was not randomized. The investigators were not blinded to allocation during the study and outcome assessment. We excluded participants who had a pre-existing cancer diagnosis, were lost to follow-up, had missing dietary assessment information before cancer onset, or were missing information for more than 80% of covariates to investigate the associations between diet scores and risk of overall and 22 specific cancers. For the analysis of identifying metabolites associated with overall cancer, from the original pool of 168 metabolites. we excluded participants who had a pre-existing cancer diagnosis, were lost to follow-up, used lipid-lowering drugs before blood collection, and those with missing data on baseline metabolites. In addition, we excluded participants lacking information on more than 80% of the basic confounding factors. All replication attempts were successful.

## Reporting summary

Further information on research design is available in the Nature Portfolio Reporting Summary linked to this article.

## Data availability

The dataset analyzed during this study is available in the UK Biobank (https://www.ukbiobank.ac.uk/) under application number 675116. The raw UK Biobank data are protected and are not available due to data privacy laws; Access can be obtained by data application for the UK Biobank platform. Source data supporting all our findings (Figs. 1–4 and Supplementary Fig. 1–5) are provided with this publication as a Source Data file. Source data are provided in this paper. Source data are provided with this paper.

## Code availability

Software used for analysis includes R v4.1.1, Python v3.11. Analysis codes used in this manuscript can be found at https://doi.org/10.5281/zenodo.10953469[76].

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

## Acknowledgements

This research was conducted using the UK Biobank Resource (https://www.ukbiobank.ac.uk/) under application number 675116. This study was supported by the National Nature Science Foundation of China (No.82373564 [L.F.Q], No.721990083 [H.S.D] and No.52208092 [H.S.D]), the Central government-led local science and technology development special project (No. 2019L3006 [H.Z.J] and 2020L3009 [H.Z.J]), the Scientific and Technological Innovation Joint Capital Projects of Fujian Province (No.2020Y9018 [H.Z.J]).

We extend our sincere appreciation to Susan Olivier for her invaluable assistance and professional insights during the writing of this paper. Her contributions significantly enhanced the readability and clarity of expression.

## Author contributions

The authors' responsibilities were as follows: Fan Y is the first author, who designed the study, analyzed the data, and wrote the manuscript; Liu FQ, Hu ZJ, and Huang SD are all senior authors on this paper, and contributed to the design of the study and revision of this paper; Hu CC, Xie XX helped to revise the manuscript for important intellectual content and contributed to the design of the study; Weng YF, Chen C, Wang ZK, He XQ and Jiang DX assisted with the data analysis. All authors read and approved the final manuscript.

## Competing interests

The authors declare no competing interests.
