## [Peer Review File · Nature Communications]

Effects of Diets on Risks of Cancer and the Mediating role of MetabolitesREVIEWER COMMENTS

Reviewer #1 (Remarks to the Author):

This is a prospective study investigating the association of two diet patterns, MedDiet and MIND diet, on risk of cancer in UKBB and testing whether these associations are mediated by metabolomic biomarkers. The study is in general very well conducted, but I am describing below some suggestions for improvement.

1. In first paragraph of Introduction, the authors cite the WCRF 3rd Expert Report. Naming IARC as part of this work is not accurate.
2. The analysis involving metabolomics was conducted using an earlier dataset in UKBB involving ~120K participants, but since more than 6 months ago, the participants with available metabolomic data has increased to ~250K, and the authors should use this dataset, which will provide better power for their metabolomic investigations.
3. The authors nicely used the UKBB 24-hour diet recalls for their dietary assessment. The majority of the participants have one such assessment available and some have multiple assessments. Relying on one such assessment may lead to misclassification as dietary intake in one day will not reflect accurately dietary habits. The authors used the average dietary assessment for participants with more than one assessment, but still the majority had only one assessment. The authors should be clear about this issue, provide relevant participant numbers, and conduct sensitivity analyses where only participants with at least two assessments are used. I consider this analysis of critical importance to assure diet misclassification is not majorly affecting the current findings.
4. The authors have adjusted for smoking status, but several of the evaluated cancers have a strong association with smoking; thus smoking intensity should be added for these models.
5. In the results, when authors describe interaction results, I noticed that in several instances they suggest interaction, but the CIs overlap in the respective models. Could this issue be double-checked please and suggest to add interaction p-values somewhere in the relevant text.

Reviewer #2 (Remarks to the Author):

The authors report on a comprehensive analysis in the UK Biobank, investigating associations of two hypothesis-driven dietary patterns and cancer incidence (overall and at 22 sites). As a second objective they investigate a potential mediating effect of a range of metabolites. The authors find both, inverse associations between adherence to both dietary patterns for overall cancer incidence and at a number of cancer sites, and a strong mediation effect by 10 metabolites.

This is an interesting and well-done analysis, whereby the analysis of the MedDiet with cancer seems rather confirmative, though the MIND Diet part as well as the mediation analysis by metabolites adds novelty. The approach is described in detail, particularly taking into account the extensive supplement, so that replication by others is facilitated.

There are two aspects which I think deserve somewhat more room in the current work, i.e. a) the selection of the metabolites and the hypotheses for their interlinkage with diet and cancer, and b) the comparability or differentiation of the two dietary patterns. Please find more on these point and some minor aspects below.

- Line 5: "Effect" or better "association"?
- Line 8: "and" the Mediterranean-DASH...?
- Line 10: Is "UKB" a common abbreviation for UK Biobank? I suggest keeping the full phrase.
- Lines 11-12: It is unclear to the reader here, if you used MEDAS as an instrument. Reading the paper I understood you worked with data from the Oxford WebQ as the dietary assessment instrument and used the screener to categorize to the MedDiet. This is unclear here. You might want to state a) the assessment instrument, and b) how you constructed the scores.
- In addition, this comment is for the whole manuscript, can you reflect somewhere about MEDAS as a basis for the MedDiet? Other studies might have categorized based on a different approach?
- Line 14: can you add follow-up time?
- Line 15: can you add how these metabolites were assessed? Or on which basis they were chosen for analysis?

- Line 77-78: not sure what you mean by "animal epidemiological" studies? And, difference to the MedDiet would be interesting to state.
- Line 87: France?
- Line 89: Analysis? "up"? Please clarify.
- Line 92: You might want to argue why they are supposed to be different from MedDiet, please compare my comment above
- Line 96: "wanted" or "warranted"?
- Line 106: You might want to add some hypotheses here? This would substantiate your choice of metabolites, I assume. The tumour microenvironment highlighted in the sentence before, how does this link to the metabolites? As the sentence reads, it does not?
- Line 109: It is unclear what "NMR metabolomic biomarker for overall cancer" are. You might want to introduce these as well in the introduction.
- Line 126: Biomarkers: which ones? Why were they assessed in the UKBiobank study? Please add some information (compare to my comment at the end of the introduction)
- Line 131: Why only 35k participants here?
- Line 155: What do you mean by "average dietary assessment"? Is this the "average dietary intake", i.e. mean of all values per participant?
- Line 158: As far as I understand, you used the Oxford WebQ to assess intake, but used the MEDAS scoring method. If I understood correct, you might want to state this clearly. I assume, if participants would answer the screening instrument themselves, this would result in a different classification.
- Line 174: Was this a targeted assessment? How were the metabolites chosen?
- Line 184: Do you mean "dietary energy intake"?
- Line 185: Do I understand correct, the touch-screen questionnaire was used for dietary energy intake? How does this assessment link with the Oxford WebQ?
- Line 210: What do you mean by "level"?
- Line 214-215: I do not understand these numbers. Above you state that you have about 35k individuals for the analysis including NMR metabolites? Here, it seems you have about 75k individuals?
- Line 236: "based on baseline dietary assessment": please clarify: this was a single Oxford WebQ?
- Line 295: "strong": are these selected because HR were lowest, or is there a definition of what is a "strong" association? You might also want to change to "association" here.
- Line 307: Was this driven by some specific cancer sites?
- Line 319: "Data"?
- Line 329: it would be really helpful for the reader to understand up front how the biomarkers for NMR were selected, or if this was an untargeted assessment?
- Line 371: "data"?
- Line 413: It would be interesting to know how the two dietary types differ and if the difference in findings is in fact attributed to the differences in the score composition or if it might also be based on different approaches to compiling the scores, i.e. using a MEDAS tool as source vs. food groups for MedDiet? Low SFA, abundant antioxidants and fiber would also qualify for MedDiet, I assume.
- Line 416-418: This seems repeated. I recommend a discussion about the differences and commonalities of the two patterns as described above.
- Line 436: how do omega-3-fatty acids relate to the patterns? These were not highlighted above as nutrients relevant to the patterns, no?
- Line 471: what is meant by "sofrito"?
- Line 474: olive "oil"?
- Line 488: Is there any evidence that there is a score trajectory in your data?
- Table 1: average household income. What are these values? How did you define "occasional drinkers"? Physical activity "per week"? "Dietary energy intake"?

Responds to reviewer 1 #1

General Comment

This is a prospective study investigating the association of two diet patterns, MedDiet and MIND diet, on risk of cancer in UKB and testing whether these associations are mediated by metabolomic biomarkers. The study is in general very well conducted, but I am describing below some suggestions for improvement.

Responses: We sincerely appreciate your time and effort on critically evaluating our manuscript. Your insightful comments and constructive feedback have proven invaluable in enhancing the quality and clarity of our work. According to your comment, we have updated the results section and corresponding description using the updated UKB dataset as you suggested. The results in the revised manuscript generally align with the initial edition. Additionally, we have thoroughly addressed each of your comments.

[1] Comment 1

In first paragraph of Introduction, the authors cite the WCRF 3rd Expert Report. Naming IARC as part of this work is not accurate.

Responses: Thank you for pointing this out. We have removed the misquoted content in the revised manuscript.

The changes can be found in “Introduction section” (*Line 61-63, page 3*) as follows:

The 3rd Expert Report of the World Cancer Research Fund (WCRF) expressed a consensus that various nutritional factors may have a “probable” causal, or protective association with cancer risk (1).

Reference:

1. WCRF. (Diet, nutrition, physical activity and cancer: a global perspective.) (2018).

[2] Comment 2

The analysis involving metabolomics was conducted using an earlier dataset in UKBB involving ~120K participants, but since more than 6 months ago, the

participants with available metabolomic data has increased to ~250K, and the authors should use this dataset, which will provide better power for their metabolomic investigations.

Responses: We have updated the NMR dataset as you suggested, and revised all the analyses and results of metabolomic investigations in the revised manuscript.

As specified on the official UK Biobank website (<https://biobank.ndph.ox.ac.uk/showcase/label.cgi?id=220>), we currently have access to both Phase 1 and Phase 2 data. This dataset includes measurements of the original 168 metabolic biomarkers obtained from EDTA plasma samples collected from approximately 280,000 participants in the UK Biobank.

The top 10 identified metabolites before and after data updating are listed in **Table 1**, and metabolites identified in both analyses are marked in bold in the table. After updating the data, the results are very similar; however, there are slight differences in a few identified metabolites and the rank of importance of the metabolites. The 6 metabolites that were identified by both analyses exhibited consistency in the associations with overall cancer risk as listed in **Table 2**.

Table 1. Identified metabolites before and after data updating

Identified metabolites after data updating	Identified metabolites before data updating
Total Lipids in VLDL	Average Diameter for HDL Particles
Total Cholines	Total Cholines
Omega-3 Fatty Acids	Omega-3 Fatty Acids
Tyrosine	Tyrosine
Glucose	Glucose
Citrate	Citrate
Creatinine	Free Cholesterol in Very Small VLDL
Albumin	Free Cholesterol in IDL
Free Cholesterol in IDL	Triglycerides in Medium LDL
Total Lipids in Large HDL	Phospholipids in Small LDL

Table 2. Associations between the 6 metabolites with overall cancer risk after and before data updating in test set

Overlapping metabolites	HR: 95%CI (Associations after date updating)	HR: 95%CI (Associations before data updating)
Total Cholines	0.921 (0.904-0.939)	0.881 (0.852-0.912)
Omega-3 Fatty Acids	0.880 (0.864-0.897)	0.829 (0.801-0.858)
Tyrosine	0.935 (0.918-0.952)	0.938 (0.907-0.970)
Glucose	0.955 (0.938-0.972)	0.924 (0.896-0.953)
Citrate	0.902 (0.885-0.918)	0.868 (0.839-0.898)
Free Cholesterol in IDL	0.907 (0.890-0.924)	0.865 (0.835-0.896)

Though there are slight differences in the results of mediation analyses, the mediating roles of the 6 identified metabolites from both analyses remain consistent. Total cholines, omega-3 fatty acids, and free cholesterol in IDL were found to play a significant role, both independently and jointly, in mediating the association of diet adherence with overall cancer risk (**Table S12** in revised Supplementary Materials and **Figure 4** in revised manuscript).

We updated **Table 2**, **Figure3-4** in the revised manuscript and **Figures S1, S4, S5** in the revised Supplementary Materials. We have also revised the corresponding text in the “Method Section” (*Line 470-476, page 17; Line 588-608, page 21-22*) and “Results Section” (*Line 223-226, page 8; Line 232-234, page 9; Line 260-271, page 10*).

[2] Comment 3

The authors nicely used the UKB 24-hour diet recalls for their dietary assessment. The majority of the participants have one such assessment available and some have multiple assessments. Relying on one such assessment may lead to misclassification as dietary intake in one day will not reflect accurately dietary habits. The authors used the average dietary assessment for participants with more than one assessment, but still the majority had only one assessment. The authors should be

clear about this issue, provide relevant participant numbers, and conduct sensitivity analyses where only participants with at least two assessments are used. I consider this analysis of critical importance to assure diet misclassification is not majorly affecting the current findings.

Responses: Thank you for the suggestion. Among the 187,485 participants who completed at least one 24-hour dietary assessment before cancer onset included in our diet score main analysis, 75,277 (40.2%) had only a single 24-hour diet assessment. 24-hour diet recalls had been widely used in previous studies (1). However, dietary intake in one day, even with repeated 24-hour diet assessments, cannot reflect accurately dietary habits and may lead to some bias. We performed sensitivity analysis with only participants with at least two assessments as you suggested. The results of sensitivity analysis in **Table 3** (also referred as **Table S14** of the Supplementary Materials) show no significant differences with the results of our main model, indicating robustness of our results. Nevertheless, we also listed this as one of the limitations of our study in the discussion part.

Table 3. Associations between MIND/MEDAS score and the risk of overall cancer using participants with at least one dietary assessment compared with using participants with at least two dietary assessments

Diet Adherence	Main Analysis* (N=187,485)			Sensitivity analysis# (N=112,208)		
	N	Incidence	HR [95%CI]	N	Incidence	HR [95%CI]
MEDAS score						
Tertile1 (0 [<4])	77109	10842	Reference	40126	5388	Reference
Tertile2 (4 [<6])	80408	11360	0.858 (0.835, 0.881)	52308	7147	0.826 (0.796, 0.856)
Tertile3 (6 [≤11])	29968	4189	0.818 (0.789, 0.849)	19774	2659	0.781 (0.744, 0.819)
Continuous*	187485	26391	0.946 (0.938, 0.953)	112208	15194	0.933 (0.923, 0.944)
P_{trend}			<0.001			<0.001
MIND score						
Tertile1 (0 [<5.5])	71846	9936	Reference	30536	3984	Reference

Tertile2 (5.5 [<7.0])	55306	7901	0.907 (0.880, 0.934)	35249	4810	0.886 (0.849, 0.924)
Tertile3 (7.0 [≤ 13.5])	60333	8554	0.838 (0.815, 0.863)	46423	6400	0.808 (0.776, 0.842)
Continuous*			0.957 (0.951, 0.963)			0.947 (0.938, 0.956)
P_{trend}			<0.001			<0.001

Footnote: *Participants completed at least one 24-hour dietary assessment before cancer onset.

Participants completed at least two 24-hour dietary assessment before cancer onset.

HR adjusted for energy, sex, completed full time education, average total household income, Townsend deprivation index, family history of cancer, smoking status, alcohol drinking status, physical activity, sleep duration, BMI, WHR, and SBP

We have added **Table S14** in the revised Supplementary Materials and also revised the corresponding part in “Discussion Section” (Line 425-432, page 15) in the revised manuscript as follows:

We used an average diet score which can better capture participants real dietary habits in our model. However, 75,277 (40.2%) participants only had a single assessment as their baseline, potentially deviating from their true dietary patterns and introducing some bias into our results. Nevertheless, sensitivity analysis including participants with at least two dietary assessments showed consistency with our main model, indicating the robustness our findings. However, it is crucial to acknowledge that a limitation still exists due to potential for bias in self-reported dietary information.

Reference:

1. Hu, FB. et al. Dietary fat and coronary heart disease: a comparison of approaches for adjusting for total energy intake and modeling repeated dietary measurements. American journal of epidemiology 149, 531-540 (1999).

[4] Common 4

The authors have adjusted for smoking status, but several of the evaluated cancers have a strong association with smoking; thus smoking intensity should be added for these models.

Responses: We appreciate your suggestion about adjusting for smoking intensity. Several cancers, i.e., head and neck cancer, esophageal cancer, and lung cancer, have been reported to have strong associations with smoking. In addition, alcohol consumption has also been recognized as a risk factor of cancer (1). Therefore, we have included not only smoking but also drinking habits as covariates in the study. Taking your advice, we have incorporated smoking intensity (pack years of smoking) and alcohol intake frequency into our models for sensitivity analysis, specifically focusing on a subset of the participants (N=52,723) with the information for these two variables.

We found the associations between the MEDAS/MIND score and overall cancer risk in the sensitivity analysis remained consistent with the main model. For specific cancer types with strong association with smoking, such as head and neck cancer, esophageal cancer, and lung cancer, we found similar results (**Table 4**). The minor differences in *HR* values are likely attributable to the fluctuations in sample size.

Table 4. Associations between tertiles of MEDAS/ MIND scores with the risk of overall and 3 specific cancers in the main and sensitivity analysis

	Main Analysis (N=187,485)			Sensitivity analysis (N=52,723)		
	N	Incidence	HR * (95% CI)	N	Incidence	HR # (95% CI)
Overall cancer						
MEDAS score						
Tertile1 (0 [<4])	77109	10842	Reference	22587	3761	Reference
Tertile2 (4 [<6])	80408	11360	0.858 (0.835, 0.881)	12267	1959	0.867 (0.820,0.916)
Tertile3 (6 [≤ 11])	29968	4189	0.818 (0.789, 0.849)	17869	2982	0.882 (0.839,0.928)
MIND score						
Tertile1 (0 [<5.5])	71846	9936	Reference	21461	3529	Reference

Tertile2 (5.5 [<7.0])	55306	7901	0.892 (0.866, 0.919)	15338	2558	0.901 (0.856,0.948)
Tertile3 (7.0 [≤ 13.5])	60333	8554	0.808 (0.784, 0.832)	15924	2615	0.825 (0.783,0.87)
Head and neck cancer						
MEDAS score						
Tertile1 (0 [<4])	66411	144	Reference	18891	65	Reference
Tertile2 (4 [<6])	69184	136	0.844 (0.665, 1.072)	10341	33	0.853 (0.559,1.302)
Tertile3 (6 [≤ 11])	25845	66	1.093 (0.809, 1.478)	14941	54	0.944 (0.649,1.374)
MIND score						
Tertile1 (0 [<5.5])	62050	140	Reference	18003	71	Reference
Tertile2 (5.5 [<7.0])	47512	107	0.925 (0.718, 1.193)	12820	40	0.716 (0.484,1.058)
Tertile3 (7.0 [≤ 13.5])	51878	99	0.752 (0.576, 0.981)	13350	41	0.652 (0.439,0.968)
Esophagus cancer						
MEDAS score						
Tertile1 (0 [<4])	66420	153	Reference	18912	86	Reference
Tertile2 (4 [<6])	69165	117	0.732 (0.573, 0.936)	10335	27	0.566 (0.366,0.876)
Tertile3 (6 [≤ 11])	25816	37	0.680 (0.470, 0.981)	14930	43	0.694 (0.476,1.014)
MIND score						
Tertile1 (0 [<5.5])	62050	140	Reference	18008	86	Reference
Tertile2 (5.5 [<7.0])	47493	88	0.805 (0.615, 1.053)	12828	27	0.880 (0.611,1.268)
Tertile3 (7.0 [≤ 13.5])	51858	79	0.687 (0.517, 0.913)	13341	43	0.593 (0.388,0.906)
Lung cancer						
MEDAS score						
Tertile1 (0 [<4])	66267	485	Reference	19194	368	Reference
Tertile2 (4 [<6])	69048	411	0.784 (0.686, 0.897)	10474	166	0.794 (0.659,0.956)
Tertile3 (6 [≤ 11])	25779	131	0.684 (0.560, 0.834)	15108	221	0.771 (0.648,0.918)
MIND score						
Tertile1 (0 [<5.5])	62369	459	Reference	18288	356	Reference
Tertile2 (5.5 [<7.0])	47716	311	0.854 (0.739, 0.988)	13012	232	0.896 (0.758,1.060)
Tertile3 (7.0 [≤ 13.5])	52036	257	0.658 (0.562, 0.771)	13476	167	0.637 (0.526,0.771)

Footnotes: HR* adjusted for energy, sex, completion of full-time education, average total household income, Townsend deprivation index, family history of cancer, smoking status, alcohol drinking status, physical activity, sleep duration, BMI, WHR, and SBP; HR# additionally adjusted for smoking intensity and alcohol intake frequency

In conclusion, making additional adjustments for smoking intensity has a limited impact on the results for both overall cancer and specific cancers. The inclusion of the smoking intensity in the model resulted in a reduction of the sample size. Therefore, the adjustment for smoking intensity was presented as sensitivity analysis in the revised manuscript.

We have added **Table S19** in the revised Supplementary Materials and also revised the corresponding part in “Method Section” (*Line 627-629, page 22*) and “Result Section” (*Line 289, page 11*) of the revised manuscript.

Reference:

1. Rumgay, H., Murphy, N., Ferrari, P., Soerjomataram, I. Alcohol and Cancer: Epidemiology and Biological Mechanisms. *Nutrients* 13, (2021).

[5] Comment 5

In the results, when authors describe interaction results, I noticed that in several instances they suggest interaction, but the CIs overlap in the respective models. Could this issue be double-checked please and suggest to add interaction p-values somewhere in the relevant text.

Responses: We have double-checked the interaction results. We have added *p*-values in the revised manuscript. For your convenience, we also present the significant interaction results for specific cancers in **Table 5** (also referred as **Table S9** of the revised Supplementary Materials). As shown in the table, despite some overlap in CIs, statistical significance for the interactions still exists. Generally, when comparing two parameter estimates, it is consistently true that non-overlapping CIs indicate statistically significant differences; however, the converse is not necessarily valid (1). Details of the explanation are shown in **Figure 1**.

Table 5. Association of MEDAS/ MIND scores with the risk of specific cancers stratified by basic characteristics

Diet score	Group	HR (95% CI)	P _{interaction}
Stomach cancer			
Smoking status			
MEDAS score	No	0.996 (0.734, 1.351)	0.012
MEDAS score	Yes	0.626 (0.467, 0.839)	
MIND score	No	1.040 (0.803, 1.347)	0.040
MIND score	Yes	0.774 (0.615, 0.975)	
Colorectal cancer			
Sex			
MEDAS score	Female	0.95 (0.862, 1.046)	0.013
MEDAS score	Male	0.816 (0.746, 0.892)	
MIND score	Female	0.965 (0.889, 1.049)	0.006
MIND score	Male	0.842 (0.782, 0.906)	
Smoking status			
MEDAS score	No	0.773 (0.702, 0.851)	<0.001
MEDAS score	Yes	0.976 (0.893, 1.067)	
MIND score	No	0.830 (0.767, 0.899)	0.007
MIND score	Yes	0.952 (0.883, 1.027)	
Liver cancer			
BMI			
MEDAS score	<25 kg/m ²	1.172 (0.782, 1.757)	0.009
MEDAS score	≥25 kg/m ²	0.694 (0.536, 0.899)	
Pancreas cancer			
Sex			
MIND score	Female	1.052 (0.89, 1.244)	0.019
MIND score	Male	0.760 (0.644, 0.896)	
Lung cancer			

		Smoking status	
MIND score	No	0.951 (0.792, 1.143)	0.001
MIND score	Yes	0.736 (0.676, 0.802)	
Hodgkin lymphoma			
		Completion of full-time education	
MEDAS score	No	1.366 (0.663, 2.813)	0.024
MEDAS score	Yes	0.595 (0.309, 1.147)	
Leukemia			
		Completion of full-time education	
MIND score	No	0.707 (0.587, 0.851)	0.046
MIND score	Yes	0.898 (0.771, 1.046)	

Figure 1. Explanation for significant difference with CI overlap in two groups

We revised the stratified analysis for associations between diet scores and overall cancer risk in “Result Section” (Line 198-201, page 7) in the revised manuscript as follows:

However, a significant interaction between MIND score with sex was observed ($P_{interaction}=0.018$), indicating a stronger association in males (HR [95%CI]: 0.880 [0.862, 0.897]) compared females (HR [95%CI]: 0.918 [0.898, 0.938]).

We also revised the stratified analysis for associations between diet scores and specific cancer risk in “Result Section” (Line 207-216, page 8) in the revised manuscript as follows:

For stomach cancer, significant interactions were observed between MEDAS and MIND scores with smoking status ($P_{interaction}$: 0.012 and 0.014 respectively); for colorectal cancer, significant interactions were observed between MEDAS and MIND score with sex ($P_{interaction}$: 0.013 and 0.006 respectively) and smoking status($P_{interaction}$: <0.001 and 0.007 respectively); for liver cancer, significant interaction between MEDAS score with BMI ($P_{interaction}$ = 0.009) was observed; for pancreas cancer, significant interaction between MIND score with sex ($P_{interaction}$ = 0.019) was observed; for lung cancer, significant interaction between MIND score with smoking status ($P_{interaction}$ = 0.001) was observed; for Hodgkin lymphoma, significant interaction between MEDAS score with completion of full-time education ($P_{interaction}$ =0.024) was observed; for leukemia, significant interaction between MIND score with completion of full-time education ($P_{interaction}$ = 0.046) was observed.

Reference:

1. Austin, PC., Hux, JE. A brief note on overlapping confidence intervals. Journal of Vascular Surgery 36, 194-195 (2002).

Responds to reviewer 2 #2

General Comment

The authors report on a comprehensive analysis in the UK Biobank, investigating associations of two hypothesis-driven dietary patterns and cancer incidence (overall and at 22 sites). As a second objective they investigate a potential mediating effect of a range of metabolites. The authors find both, inverse associations between adherence to both dietary patterns for overall cancer incidence and at a number of cancer sites, and a strong mediation effect by 10 metabolites.

This is an interesting and well-done analysis, whereby the analysis of the MedDiet with cancer seems rather confirmative, though the MIND Diet part as well as the mediation analysis by metabolites adds novelty. The approach is described in detail, particularly taking into account the extensive supplement, so that replication by others is facilitated.

There are two aspects which I think deserve somewhat more room in the current work, i.e. a) the selection of the metabolites and the hypotheses for their interlinkage with diet and cancer, and b) the comparability or differentiation of the two dietary patterns.

Please find more on these point and some minor aspects below.

Responses: We sincerely appreciate your valuable comments and suggestions. We first address your two main concerns, and then reply to your minor comments one by one. We have revised our manuscript according to your comments.

General Comment 1

The selection of the metabolites and the hypotheses for their interlinkage with diet and cancer

Responses:

(1) The Selection of Metabolite

In total, 168 original circulating metabolites, which were measured by Nuclear Magnetic Resonance (NMR) spectroscopy, were available in the UK biobank data, as has already reported in several published papers (1,2). In the process of selecting cancer

related metabolites from the original pool of 168 metabolites, we employed a sequential analysis strategy that combined a Cox regression model, elastic net model (ENM) and gradient boost model (GBM).

Details of statistics analysis for metabolites selection have been presented in the “Method Section” in revised manuscript (Line 588-608, page 21-22) as follows:

*We used a sequential analysis strategy that combined Cox regression model, ENM and GBM to select metabolites associated with overall cancer risk from the original pool of 168 NMR metabolites, in 202,303 individuals. First, all participants (n=202,303) were randomly assigned to a training (n=121,382) or a testing set (n=80,921). In the training set, a Cox regression model was used to test for individual associations between metabolites and overall cancer risk (Results are shown in **Table S28**). FDR (false discovery rate) adjusted P values were calculated using the Benjamini-Hochberg method. Then, ENM was used to select metabolites from an extensive pool of candidates following the Cox regression analysis, taking into consideration potential multicollinearity between metabolites. ENM is a supervised machine learning algorithm that seamlessly integrates least absolute shrinkage and selection operator (LASSO) regression and ridge regression. It is known for its excellent probabilistic interpretation of variables and suitability for disease prediction (3). A 10-fold cross-validation was used to select optimal λ and β coefficients for the ENM models. To achieve sparsity of the model, the crossover and overlapped metabolites identified with α value of 0.25, 0.50, 0.75, and 1.00 from the ENM models. Subsequently, we used GBM analysis to evaluate the relative importance of metabolites. Finally, the top 10 metabolites from the GBM analysis were considered as identified metabolites. The associations between these metabolites with overall and 22 specific types of cancer risks were tested by Cox regression model in the testing set.*

(2) The hypotheses for the metabolite’s interlinkage with diet and cancer

The hypotheses for the metabolite's interlinkage with diet and cancer has been added in the “Introduction Section” of the revised manuscript (Line 103-112, page 4):

A prominent feature of cancer is the induced metabolic transformation, which offers a critical target for cancer therapy (4). Dietary factors can significantly influence

tumor growth by altering cell metabolism, partly through changes in nutrient access and utilization by cancer cells (5). Changes in dietary compositions contribute to perturbations in plasma metabolite levels, subsequently affecting metabolite concentrations in the tumor microenvironment. The metabolic activity of cancer cells could be altered in response to changes of metabolite concentrations in their local environment (6). Therefore, our hypothesis is that dietary factors may influence the levels of metabolites in the tumor microenvironment, thereby modulating cancer cell metabolism and ultimately impacting tumor growth.

As for the cancer related metabolites identified and reported in our study, we also added an explanation of potential mechanisms underlying their linkage to cancer in the “Discussion Section” of the revised manuscript (Line 384-388, page 14 and Line 393-396, page 14) and highlighted the revised manuscript in yellow.

Reference:

1. Buerge, T. et al. Metabolomic profiles predict individual multidisease outcomes. *Nature Medicine* 28, 2309-2320 (2022).
2. Zhang, X. et al. Plasma metabolomic profiles of dementia: a prospective study of 110,655 participants in the UK Biobank. *BMC medicine* 20, 252 (2022).
3. Zou, H., Hastie, T. Regularization and Variable Selection Via the Elastic Net. *Journal of the Royal Statistical Society Series B: Statistical Methodology* 67, 301-320 (2005).
4. Elia, I., Haigis, MC. Metabolites and the tumour microenvironment: from cellular mechanisms to systemic metabolism. *Nat Metab* 3, 21-32 (2021).
5. Bose, S., Allen, AE., Locasale, JW. The Molecular Link from Diet to Cancer Cell Metabolism. *Mol Cell* 78, 1034-1044 (2020).
6. Lien, EC., Vander, Heiden MG. A framework for examining how diet impacts tumour metabolism. *Nature reviews Cancer* 19, 651-661 (2019).

General Comment 2

The comparability or differentiation of the two dietary patterns

Responses: As you suggested, the comparability and differentiation of the components of the two dietary patterns have been added in the “Introduction Section” (Line 68-79,

page 3), “Result Section” (Line 154-170, page 6; Line 182-193, page 7), and “Discussion Section” (Line 308-313, page 11; Line 336-348, page 12-13). The revisions are as follows:

(1) Introduction part

The MedDiet is distinguished by its high consumption of vegetables and fruits, regular intake of fish and seafood, and moderate intake of red wine (1). It has been widely recognized as one of the healthiest dietary patterns globally. Multiple studies have found that adherence to a MedDiet can reduce the risk and mortality of metabolic diseases (2-4). The MINDDiet was inspired by the renowned MedDiet, but has been modified based on the most compelling evidence from human epidemiological and animal studies in the diet-dementia field (5). The MINDDiet places a particular emphasis on natural plant-based foods, while limiting the intake of animal and high saturated fat foods. Diverging from MedDiet, the MINDDiet distinctively encourages only the specific consumption of green leafy vegetables and berries, while not explicitly addressing the intake of other types of fruits.

(2) Result part

*Associations between MEDAS and MIND scores with overall cancer risk were examined using Cox regression models, with diet scores treated as categorical and continuous variables respectively. We used 3 different models, adjusting for different covariates to capture these associations. Results from the 3 different models, adjusting for different covariates, consistently showed that higher MEDAS and MIND scores were significantly associated with a lower overall cancer risk ($P < 0.001$), as shown in **Figure 1**. When scores were treated as categorical variables (reference=tertile 1), the HR (95%CI) values for MEDAS and MIND scores at tertile 3 using model 1, adjusting for energy intake, were 0.849 (0.819, 0.879) and 0.838 (0.815, 0.863) respectively; after further adjusting for demographic factors (including sex, completion of full-time education, average total household income, Townsend deprivation index, and family history of cancer) in model 2, HR (95%CI) values were 0.802 (0.773, 0.831) and 0.795 (0.772, 0.819) respectively; similar results were observed in model 3 with additional adjustment for behavioral risk factors (including smoking status, alcohol drinking*

status, physical activity, sleep duration, BMI, WHR, and systolic blood pressure [SBP]), with HR (95%CI) values of 0.818 (0.789, 0.849), respectively. Consistent results were observed when diet scores were treated as continuous variable as shown in **Figure 1**.

For 22 specific cancers, we examined the associations using model 3, with diet scores as categorical variables. As shown in **Figure 2**, higher adherence to the MEDAS score demonstrated significant associations with reduced risks of 14 specific cancers, namely esophagus cancer, colorectal cancer, lung cancer, malignant melanoma, breast cancer, uterus and cervix cancer, prostate cancer, testis cancer, kidney cancer, bladder cancer, brain cancer, thyroid cancer, non-Hodgkin lymphoma, and leukemia. While, adherence to the MIND score exhibited inverse associations with 13 specific cancers, including head and neck cancer, esophagus cancer, colorectal cancer, lung cancer, malignant melanoma, breast cancer, uterus and cervix cancer, prostate cancer, kidney cancer, thyroid cancer, non-Hodgkin lymphoma, multiple myeloma, and leukemia. Significant protective effects of both MEDAS and MIND scores at tertile 2 and tertile 3 were observed in colorectal cancer, lung cancer, breast cancer, and prostate cancer.

(3) Discussion part

Generally, we observed significant negative associations between both MEDAS and MIND scores and the risk of overall, and several specific cancers, indicating that both dietary patterns, characterized by abundant consumption of vegetables, legumes, nuts, grains, fish, and seafood, while reducing the consumption of butter, margarine, cream, cheese, red meat and products, as well as pastries and sweets, provide protective effects against cancer.

However, we found no significant association between a MINDDiet and brain cancer, contrary to our expectations, as the MINDDiet is derived from the MedDiet with specific modifications based on evidence from neurodegenerative diseases (5). Although results from a case-control study reported inverse associations of a MINDDiet with the risk of glioma (6), evidence from our study indicated no significant protective effects of a MINDDiet on neurogenic neoplasms. Compared to the MINDDiet, a MedDiet exhibited a stronger protective effect against several specific cancer types, including kidney cancer, brain cancer, and thyroid cancer. This was mainly due to the

differences in the composition of the two diets: MINDDiet explicitly specifies the consumption of green leafy vegetables and berries, while MedDiet also includes other types of fruits and emphasizes moderate consumption of dairy products. Another potential explanation lies in the variations between the two dietary scoring systems in terms of the specified quantities for various components.

Reference:

1. Ratjen, I. et al. Postdiagnostic Mediterranean and Healthy Nordic Dietary Patterns Are Inversely Associated with All-Cause Mortality in Long-Term Colorectal Cancer Survivors¹². *The Journal of Nutrition* 147, 636-644 (2017).
2. Grosso, G. et al. A comprehensive meta-analysis on evidence of Mediterranean diet and cardiovascular disease: Are individual components equal? *Crit Rev Food Sci Nutr* 57, 3218-3232 (2017).
3. Schwingshackl, L., Schwedhelm, C., Galbete, C., Hoffmann, G. Adherence to Mediterranean Diet and Risk of Cancer: An Updated Systematic Review and Meta-Analysis. *Nutrients* 9, (2017).
4. Sofi, F., Abbate, R., Gensini, GF., Casini, A. Accruing evidence on benefits of adherence to the Mediterranean diet on health: an updated systematic review and meta-analysis. *Am J Clin Nutr* 92, 1189-1196 (2010).
5. Morris, MC. et al. MIND diet slows cognitive decline with aging. *Alzheimers Dement* 11, 1015-1022 (2015).
6. Soltani, S. et al. Adherence to the MIND diet in relation to glioma: a case-control study. *Nutritional neuroscience* 25, 771-778 (2022).

Minor Comments

[1] Comment 1

“Effect” or better “association”

Response: We used “association” instead of “effect” in the revised manuscript as you suggested.

[2] Comment 2

Line 8: “and” the Mediterranean-DASH...?

Response: We have revised it to “a Mediterranean-DASH Diet Intervention for Neurodegenerative Delay diet (MINDDiet)” in “Abstract section” (*Line 33-34, page 2*)

[3] Comment 3

Line 10: Is “UKB” a common abbreviation for UK Biobank? I suggest keeping the full phrase.

Response: Yes. We have used the full phrase "UK Biobank" instead of “UKB” as you suggested.

[4] Comment 4

Lines 11-12: It is unclear to the reader here, if you used MEDAS as an instrument. Reading the paper I understood you worked with data from the Oxford WebQ as the dietary assessment instrument and used the screener to categorize to the MedDiet. This is unclear here. You might want to state a) the assessment instrument, and b) how you constructed the scores.

Response: Sorry for the confusion. The UK Biobank study collected participants’ 24-hour dietary information using Oxford WebQ. We then utilized the MedDiet Adherence Screener (MEDAS) scoring and MIND scoring methods to quantify the adherence to Meddiet and MINDDiet, respectively, based on the dietary information from Oxford WebQ (1). Specifically, for each food component, different scores are assigned based on the intake level for the two scoring methods as shown in **Tables S25, S26** of the revised Supplementary Material. The sum of all component scores constitutes the overall dietary scores (2,3).

Due to the word limits for the abstract (150 words), we have stated the assessment instrument and how we constructed the scores in the “Methods Section” in the revised manuscript (*Line 514-536, page 18-19*).

Reference:

1. Greenwood, DC. et al. Validation of the Oxford WebQ Online 24-Hour Dietary Questionnaire Using Biomarkers. American journal of epidemiology 188, 1858-1867 (2019).
2. Shannon, OM. et al. Mediterranean diet adherence is associated with lower dementia risk,

independent of genetic predisposition: findings from the UK Biobank prospective cohort study. BMC medicine 21, 81 (2023).

3. Cornelis, MC., Agarwal, P., Holland, TM., van, Dam RM. MIND Dietary Pattern and Its Association with Cognition and Incident Dementia in the UK Biobank. Nutrients 15, (2022).

[5] Comment 5

In addition, this comment is for the whole manuscript, can you reflect somewhere about MEDAS as a basis for the MedDiet? Other studies might have categorized based on a different approach?

Response: We have included an explanation about using MEDAS to quantify MedDiet adherence in the “Method Section” (Line 514-520, page 18) as follows, and revised the corresponding expression for the whole manuscript.

We used the MEDAS scoring method to quantify adherence to the MedDiet based on the dietary assessment recorded by Oxford WebQ. The MEDAS scoring method originated from the Prevención con Dieta Mediterránea (PREDIMED) trial, and has been widely used in both trials and observational studies (1-2). MEDAS has been validated in a prospective study by Gregory et.al conducted in the UK (3), has received endorsement from the American Heart Association, and has become a rapid dietary assessment screening tool in clinical practice, due to its reliability and practical utility (4).

Reference:

1. Siervo, M., Shannon, OM., Llewellyn, DJ., Stephan, BC., Fontana, L. Mediterranean diet and cognitive function: From methodology to mechanisms of action. Free Radic Biol Med 176, 105-117 (2021).

2. Shannon, OM. et al. Mediterranean diet adherence and cognitive function in older UK adults: the European Prospective Investigation into Cancer and Nutrition-Norfolk (EPIC-Norfolk) Study. The American journal of clinical nutrition 110, 938-948 (2019).

3. Gregory, S., Ritchie, CW., Ritchie, K., Shannon, O., Stevenson, EJ., Muniz-Terrera, G. Mediterranean diet score is associated with greater allocentric processing in the EPAD LCS cohort: A comparative analysis by biogeographical region. Front Aging 3, 1012598 (2022).

4. Vadiveloo, M. et al. Rapid Diet Assessment Screening Tools for Cardiovascular Disease Risk Reduction Across Healthcare Settings: A Scientific Statement From the American Heart Association. *Circ Cardiovasc Qual Outcomes* 13, e000094 (2020).

[6] Comment 6

Line 14: can you add follow-up time?

Response: We have added the description of follow-up time as you suggested in “Abstract” in the revised manuscript. Due to the word limits for the abstract (150 words), details of follow-up time have been included in the “Result Section”.

(1) Abstract part (Line 31-35, page 2)

Based on 187,485 participants in UK Biobank with follow-up time up to 15.7 years, we investigated the associations between adherence to a Mediterranean diet (MedDiet) or a Mediterranean-DASH Diet Intervention for Neurodegenerative Delay diet (MINDDiet) and the risk of overall, and 22 specific cancers.

(2) Result part (Line 129-132, page 5)

Over the follow-up period, spanning from 0.1 to 15.7 years, for each participant (median = 13.2 years), a total of 26,391 cancer cases were diagnosed, with 12,437 (47.1%) occurring in females and 13,954 (52.9%) in males.

[7] Comment 7

Line 15: can you add how these metabolites were assessed? Or on which basis they were chosen for analysis?

Response: We used a sequential analysis strategy that combined a Cox regression model, elastic net model (ENM) and gradient boost model (GBM), and identified 10 metabolites from the original pool of 168 metabolites. A detailed description has been provided in the response to your “General Comment 1”, and has been added in the “Method Section” of our revised manuscript (*Line 588-608, page 21-22*).

[8] Comment 8

Line 77-78: not sure what you mean by "animal epidemiological" studies? And,

difference to the MedDiet would be interesting to state.

Response: Sorry for typographical error and we have revised it to “animal studies” in the “Introduction Section” in the revised manuscript (*Line 75, page 3*).

The differences between MedDiet and MINDDiet can be found in the reply to your “General Comment 2”, and has been added in the “Introduction Section” (*Line 68-79, page 3*), “Result Section” (*Line 154-170, page 6; Line 182-193, page 7*), and “Discussion Section” (*Line 308-313, page 11; Line 336-348, page 12-13*) of our revised manuscript.

[9] Comment 9

Line 87: France?

Response: Yes, it is France. Sorry for the typing mistake, and we have revised (*Line 88, page 4*).

[10] Comment 10

Line 89: Analysis? “up”? Please clarify.

Response: Thank you for your careful review and suggestions. We have removed “up” and revised “encompassing 117 studies up” to “encompassing 117 studies” (*Line 90, page 4*).

[11] Comment 11

Line 92: You might want to argue why they are supposed to be different from MedDiet, please compare my comment above

Response: MedDiet was distinguished by its high consumption of vegetables and fruits, regular intake of fish and seafood, and moderate intake of red wine (1). It was widely recognized as one of the healthiest dietary patterns globally. However, the MINDDiet placed a particular emphasis on natural plant-based foods while limiting the intake of animal and high-saturated fat foods. Remarkably, the MINDDiet distinctively encouraged the consumption of green leafy vegetables and berries, while not explicitly addressing the intake of other types of fruits (2). We have added this in “Introduction

Section” (Line 68-79, page 3) of the revised manuscript.

Reference:

1. Ratjen, I. et al. Postdiagnostic Mediterranean and Healthy Nordic Dietary Patterns Are Inversely Associated with All-Cause Mortality in Long-Term Colorectal Cancer Survivors. *The Journal of Nutrition* 147, 636-644 (2017).
2. Morris, MC. et al. MIND diet slows cognitive decline with aging. *Alzheimers Dement* 11, 1015-1022 (2015).

[12] Comment 12

Line 96: “wanted” or “warranted”?

Response: We rephrased the sentence using “warranted” instead of “wanted” (Line 96, page 4).

[13] Comment 13

Line 106: You might want to add some hypotheses here? This would substantiate your choice of metabolites, I assume. The tumour microenvironment highlighted in the sentence before, how does this link to the metabolites? As the sentence reads, it does not?

Response: We have added a description of the potential hypotheses between metabolites and cancer in the “Introduction Section” (Line 103-112, page 4) as follows:

A prominent feature of cancer is the induced metabolic transformation, which offers a critical target for cancer therapy (1). Dietary factors can significantly influence tumor growth by altering cell metabolism, partly through changes in nutrient access and utilization by cancer cells (2). Changes in dietary compositions contribute to perturbations in plasma metabolite levels, subsequently affecting metabolite concentrations in the tumor microenvironment. The metabolic activity of cancer cells could be altered in response to changes of metabolite concentrations in their local environment (3). Therefore, our hypothesis is that dietary factors may influence the levels of metabolites in the tumor microenvironment, thereby modulating cancer cell metabolism and ultimately impacting tumor growth.

More details can be found in the reply to your “General Comment 1”.

Reference:

1. Elia, I., Haigis, MC. Metabolites and the tumour microenvironment: from cellular mechanisms to systemic metabolism. *Nat Metab* 3, 21-32 (2021).
2. Bose, S., Allen, AE., Locasale, JW. The Molecular Link from Diet to Cancer Cell Metabolism. *Mol Cell* 78, 1034-1044 (2020).
3. Lien, EC., Vander, Heiden MG. A framework for examining how diet impacts tumour metabolism. *Nature reviews Cancer* 19, 651-661 (2019).

[14] Comment 14

Line 109: It is unclear what "NMR metabolomic biomarker for overall cancer" are. You might want to introduce these as well in the introduction.

Response: Sorry for the confusion. “NMR metabolomic biomarkers for overall cancer” refers to the NMR metabolites which are identified to be associated with overall cancer risk from all original measured nuclear magnetic resonance (NMR) metabolites. We have clarified this in “Introduction Section” (Line 121-124, page 5).

[15] Comment 15

Line 126: Biomarkers: which ones? Why were they assessed in the UKBiobank study? Please add some information (compare to my comment at the end of the introduction)

Response: “Biomarkers” referred to the metabolites identified to be associated with overall cancer risk from all original measured nuclear magnetic resonance (NMR) metabolites in the UK Biobank data (Line 121-124, page 5). We have explained why NMR metabolites are measured in the UK Biobank (Line 112-118, page 4-5) study and why we identified these “biomarkers” in “Introduction Section” (Line 103-112, page 4).

The revision is as follows:

(1) Why NMR metabolites are measured in the UK Biobank study

Metabolomics enables the comprehensive analysis of small molecules (metabolites) in biological samples, thereby offering valuable insights into the dynamic metabolic processes influenced by diet (1). In the field of metabolomics, Nuclear Magnetic

Resonance (NMR) spectroscopy has become one of the primary analytical methods due to its notable advantages such as high reproducibility and quantitative capabilities, non-selective and non-invasive nature, and the ability to identify unknown metabolites in complex mixtures (2).

(2) Why we identified these “biomarkers”

A prominent feature of cancer is the induced metabolic transformation, which offers a critical target for cancer therapy (3). Dietary factors can significantly influence tumor growth by altering cell metabolism, partly through changes in nutrient access and utilization by cancer cells (4). Changes in dietary compositions contribute to perturbations in plasma metabolite levels, subsequently affecting metabolite concentrations in the tumor microenvironment. The metabolic activity of cancer cells could be altered in response to changes of metabolite concentrations in their local environment (5). Therefore, our hypothesis is that dietary factors may influence the levels of metabolites in the tumor microenvironment, thereby modulating cancer cell metabolism and ultimately impacting tumor growth.

Reference:

1. Langenau, J. et al. Blood Metabolomic Profiling Confirms and Identifies Biomarkers of Food Intake. *Metabolites* 10, (2020).
2. Nagana, Gowda GA., Raftery, D. NMR-Based Metabolomics. *Adv Exp Med Biol* 1280, 19-37 (2021).
3. Elia, I., Haigis, MC. Metabolites and the tumour microenvironment: from cellular mechanisms to systemic metabolism. *Nat Metab* 3, 21-32 (2021).
4. Bose, S., Allen, AE., Locasale, JW. The Molecular Link from Diet to Cancer Cell Metabolism. *Mol Cell* 78, 1034-1044 (2020).
5. Lien, EC., Vander, Heiden MG. A framework for examining how diet impacts tumour metabolism. *Nature reviews Cancer* 19, 651-661 (2019).

[16] Comment 16

Line 131: Why only 35k participants here?

Response: In the initial version of our manuscript, though the total number of the study

population was 176,573, we did the mediation analysis with only 35k participants due to the exclusion of participants with missing diet assessment or NMR metabolite information. Actually, due to variations in the availability of participants' variables information, the numbers of participants differed in analyses for “the association between diet score and overall cancer”, “the association between NMR metabolites and overall cancer” and “the mediation effect of the NMR metabolites on the association between diet score and overall cancer”.

In the revised manuscript, according to Comment 2 of reviewer 1, we used the new UK Biobank data updated 6 months ago, and therefore had a larger study population. Consequently, the number of participants for mediation analysis increased to 85,669. Accordingly, we have updated the number from 35k to 85k in the revised manuscript. We included a flowchart of our study as **Figure 1** (also referred to **Figure S6** in the revised Supplementary Materials) for your better understanding.

It should be pointed out that, though the populations varied for each of the analyses, the basic characteristic distributions of the populations were very similar (see **Table 1** in the revised manuscript, and **Tables S4-S6** in the revised Supplementary Materials for more details).

Figure 1 Flowchart of the study

[17] Comment 17

Line 155: What do you mean by “average dietary assessment”? Is this the “average dietary intake”, i.e. mean of all values per participant?

Response: Yes, “average dietary assessment” refers to the cumulative average of dietary intake calculated from all of the preceding dietary measurements. Specifically, we calculated the dietary score for each of the 24-hour dietary assessments using the Oxford WebQ, and then calculated the average dietary score of all assessments for each participant. We also have supplemented the corresponding description in the “Methods Section” in the revised manuscript (*Line 509-511, page 18*).

[18] Comment 18

Line 158: As far as I understand, you used the Oxford WebQ to assess intake, but used the MEDAS scoring method. If I understood correct, you might want to state

this clearly. I assume, if participants would answer the screening instrument themselves, this would result in a different classification.

Response: Sorry for the confusion. We assessed dietary intake based on the information from the Oxford WebQ. We then utilized the MEDAS scoring and MIND scoring methods to quantify adherence to the Meddiet or MINDDiet, respectively, based on the dietary information from the Oxford WebQ. To express this more clearly, we have supplemented the corresponding description in the “Methods Section” in the revised manuscript (Line 513-536, page 18-19). More details can be found in the reply to your “Comment 4”.

[19] Comment 19

Line 174: Was this a targeted assessment? How were the metabolites chosen?

Response: Yes, it’s a targeted assessment and we have revised it in the “Method Section” (Line 538-540, page 19) as you suggested. We used a sequential analysis strategy of combined Cox regression model, elastic net model (ENM) and gradient boost model (GBM) to choose important metabolites from 168 metabolites. More details can be found in the reply to your “General Comment 1”.

[20] Comment 20

Line 184: Do you mean “dietary energy intake”?

Response: Yes, you are correct. We have used “dietary energy intake” in the revised manuscript.

[21] Comment 21

Line 185: Do I understand correct, the touch-screen questionnaire was used for dietary energy intake? How does this assessment link with the Oxford WebQ?

Response: Sorry for the confusion. The assessment of dietary energy intake was conducted using the Oxford WebQ, while information of covariates including age at recruitment, sex, completion of full-time education, average total household income, Townsend deprivation index, family history of cancer, smoking status, alcohol drinking,

physical activity, sleep duration, body mass index, waist-hip ratio, systolic blood pressure was collected from touch-screen questionnaire and physical measurements. We have revised the “Method Section” to reflect this (*Line 558-560, page 20*).

[22] Comment 22

Line 210: What do you mean by “level”?

Response: Level refers to education level. In order to express it more clearly, we have revised it in “Method section” using “completion of full-time education (no, yes)” (*Line 582-586, page 21*).

[23] Comment 23

Line 214-215: I do not understand these numbers. Above you state that you have about 35k individuals for the analysis including NMR metabolites? Here, it seems you have about 75k individuals?

Response: Due to variations in the availability of participants’ variables information, the numbers of populations differed in analyses for “the association between diet score and overall cancer”, “the association between NMR metabolites and overall cancer” and “the mediation effect of the NMR metabolites on the association between diet score and overall cancer”. In the initial version of our manuscript, we conducted analysis in 75k individuals to examine the association between NMR metabolites and overall cancer, and conducted mediation analysis in 35k participants for the mediation effect of NMR metabolites on the association between diet score and overall cancer.

In the revised manuscript, according to Comment 2 of reviewer 1, we used the new UK Biobank data updated 6 months ago, and so now have a larger study population. Therefore, 35k and 75k have been changed to 85k and 200k respectively in our revised manuscript.

It should be pointed out that, though the populations varied for each of the analyses, we have examined the basic characteristic distributions of these different populations and found that they were very similar. More details can be found in our reply to your comment 16, where we also include a flowchart of our study

[24] Comment 24

Line 236: “based on baseline dietary assessment”: please clarify: this was a single Oxford WebQ?

Response: Yes, you are right. “Based on baseline dietary assessment” refers to the utilization of the single 24-hour dietary information collected by Oxford WebQ when each participant entered the study.

[25] Comment 25

Line 295: “strong”: are these selected because HR were lowest, or is there a definition of what is a “strong” association? You might also want to change to “association” here.

Response: The word “strong” refers to a relatively lower HR value. We have not defined what is a “strong” association in our initial version. We don't use this word in the revised version.

[26] Comment 26

Line 307: Was this driven by some specific cancer sites?

Response: Yes, the interactions between certain covariates and dietary scores in cancer risk may be driven by the specific types of cancer. For example, for gastric cancer, colorectal cancer, and lung cancer, our study revealed, we found significant interactions between smoking and dietary scores; however, the interactions were not significant for other types of cancer.

[27] Comment 27

Line 319: “Data”?

Response: Yes, it should be data and we have revised.

[28] Comment 28

Line 413: It would be interesting to know how the two dietary types differ and if the difference in findings is in fact attributed to the differences in the score composition

or if it might also be based on different approaches to compiling the scores, i.e. using a MEDAS tool as source vs. food groups for MedDiet? Low SFA, abundant antioxidants and fiber would also qualify for MedDiet, I assume.

Response:

(1) How the two dietary types differ

MedDiet is distinguished by its high consumption of vegetables and fruits, regular intake of fish and seafood, and moderate intake of red wine (1). It was widely recognized as one of the healthiest dietary patterns globally. However, the MINDDiet placed a particular emphasis on natural plant-based foods while limiting the intake of animal and high-saturated fat foods. Remarkably, the MINDDiet distinctively encouraged the consumption of green leafy vegetables and berries, while not explicitly addressing the intake of other types of fruits (2). More details can be found in the reply to your “General Comment 2”.

(2) If the difference in findings is in fact attributed to the differences in the score composition or if it might also be based on different approaches to compiling the scores

Besides MEDAS scoring used to calculate the MedDiet adherence score in our main model, we performed sensitivity analysis using an alternative scoring approach named “MEDAS continuous scoring method” proposed by Shannon et al., which employs a linear equation ($y=ax+b$), to assign continuous scores between 0 and 1 based on the proximity to dietary targets (3). The results (**Table S28** in the revised Supplementary Materials) remain consistent, indicating that the difference is mainly due to scoring composition rather than scoring methods.

We have added this in “Sensitivity analyses” in “Method Section” (Line 623-625, page 22) and “Sensitivity analyses” in “Result Section” (Line 285-286, page 10).

Reference:

1. Ratjen, I. et al. Postdiagnostic Mediterranean and Healthy Nordic Dietary Patterns Are Inversely Associated with All-Cause Mortality in Long-Term Colorectal Cancer Survivors. *The Journal of Nutrition* 147, 636-644 (2017).
2. Morris, MC. et al. MIND diet slows cognitive decline with aging. *Alzheimers Dement* 11, 1015-

1022 (2015).

3. Shannon, OM. et al. Mediterranean diet adherence is associated with lower dementia risk, independent of genetic predisposition: findings from the UK Biobank prospective cohort study. BMC medicine 21, 81 (2023).

[29] Comment 29

Line 416-418: This seems repeated. I recommend a discussion about the differences and commonalities of the two patterns as described above.

Response: Thank you and we added a discussion about the differences and commonalities of the two patterns in the “Discussion Section” of the revised manuscript (*Line 308-313, page 11; Line 336-348, page 12*). Details can also be found in the reply to your “General comment 2”. We also addressed this in replies to “Comment 11” and “Comment 28”.

[30] Comment 30

Line 436: how do omega-3-fatty acids relate to the patterns? These were not highlighted above as nutrients relevant to the patterns, no?

Response: Both dietary patterns include foods such as fish or nuts, known for being rich sources of omega-3 fatty acids. Prior studies had also recognized omega-3 fatty acids as crucial bioactive compounds contributing to the health benefits associated with a MedDiet (1).

In our analysis of associations between specific dietary components of the two dietary patterns with cancer risk, nuts and seafood, which are abundant sources of omega-3 fatty acids, were found to be associated with a lower risk of overall cancer (**Figure S1** in revised Supplementary Materials). This indicates the benefits of omega-3 fatty acids as nutrients relevant to these dietary patterns.

We have also revised in “Discussion Section” (*Line 381-384, page 14*) as follows:

Both dietary patterns include foods rich in omega-3 fatty acids, such as fish and nuts. Prior studies have also recognized omega-3 fatty acids as crucial bioactive

compounds that contribute to the health benefits associated with a MedDiet (1).

Reference:

1. Schwingshackl, L., Morze, J., Hoffmann, G. Mediterranean diet and health status: Active ingredients and pharmacological mechanisms. *Br J Pharmacol* 177, 1241-1257 (2020).

[31] Comment 31

Line 471: what is meant by “sofrito”?

Response: “Sofrito” is a term originating from Spain which refers to traditional Mediterranean sauce that typically consists of garlic, onion, and tomatoes cooked in olive oil. It serves as a flavorful base for many Mediterranean dishes, contributing to the distinctive taste and aroma of the cuisine. We also added a short explanation in the “Discussion Section” (*Line 436-438, page 16*) of the revised manuscript as follows:

The lack of information about olive oil or sofrito (a traditional Mediterranean sauce) in the dietary assessment may introduce bias for the evaluation of MEDAS and MIND scores.

[32] Comment 32

Line 474: olive “oil”?

Response: Yes, you are correct. We have revised (*Line 440, page 16*). Thank you.

[33] Comment 33

Line 488: Is there any evidence that there is a score trajectory in your data?

Response: We apologize for the misunderstanding. We did not conduct a trajectory analysis in the present study. Here, our intention is to convey that there is a significant time gap between information collections for dietary data and cancer incidences: the latest update of dietary data in the UK Biobank only extends to 2012, while the occurrence of cancer outcomes can be traced up to 2022. Participants may have altered their dietary behavior patterns during this time period, which could introduce bias into our analysis. If there is synchronized dietary information, it can better validate the results of our study. However, as you suggested, trajectory analysis is indeed an

interesting and inspiring topic that can be explored in future studies.

The revision can be found in “Discussion Section” (Line 424-429, page 15) as follows:

There is a significant time gap between the last update for dietary and cancer incidence data. The dietary data was last updated in 2012; however, the last follow-up for incidence of cancer was conducted up until November 31, 2022 in England, August 31, 2022 in Scotland, and May 31, 2022 in Wales. This temporal misalignment could introduce some bias, as participants may have altered their dietary behavior patterns during this time period.

[34] Comment 34

Table 1: average household income. What are these values? How did you define “occasional drinkers”? Physical activity “per week”? “Dietary energy intake”?

Response:

(1) Average household income. What are these values?

“Average household income” is a continuous variable, which means the average total household income per year before tax. In our model, we transformed it into a binary variable based on the median value (<30,999£ and ≥30,999£ groups).

(2) How did you define “occasional drinkers”?

Participants who reported drinking more than once a week were categorized as regular drinkers, while those who reported drinking one to three times a month or only on special occasions were considered occasional drinkers, we added the definition in the “Method Section” (Line 554-555, page 20).

(3) Other expression concerns

Yes, we tend to mean “Physical activity per week” and “Dietary energy intake”. We have revised them in the revised manuscript.

Reviewer #1 (Remarks to the Author):

The authors have appropriately replied to the reviewer comments. I have no further comments.